# Extendable piezo/ferroelectricity in nonstoichiometric 2D transition metal dichalcogenides

Yi Hu[1,2], Lukas Rogée[1], Weizhen Wang[1], Lyuchao Zhuang[1], Fangyi Shi[1], Hui Dong[1], Songhua Cai ◎[1], Beng Kang Tay ◎[2,3] & Shu Ping Lau ◎[1] ✉

Engineering piezo/ferroelectricity in two-dimensional materials holds significant implications for advancing the manufacture of state-of-the-art multifunctional materials. The inborn nonstoichiometric propensity of two-dimensional transition metal dichalcogenides provides a spiffy ready-available solution for breaking inversion centrosymmetry, thereby conducing to circumvent size effect challenges in conventional perovskite oxide ferroelectrics. Here, we show the extendable and ubiquitous piezo/ferroelectricity within nonstoichiometric two-dimensional transition metal dichalcogenides that are predominantly centrosymmetric during standard stoichiometric cases. The emerged piezo/ferroelectric traits are aroused from the sliding of van der Waals layers and displacement of interlayer metal atoms triggered by the Frankel defects of heterogeneous interlayer native metal atom intercalation. We demonstrate two-dimensional chromium selenides nanogenerator and iron tellurides ferroelectric multilevel memristors as two representative applications. This innovative approach to engineering piezo/ferroelectricity in ultrathin transition metal dichalcogenides may provide a potential avenue to consolidate piezo/ferroelectricity with featured two-dimensional materials to fabricate multifunctional materials and distinguished multiferroic.

Owing to the advantages of low power consumption, minor consumption of input materials, substantial flexibility and high-density integration, two-dimensional (2D) piezoelectrics and ferroelectrics (piezo/ferroelectrics) are significantly expected as potential candidates in building next-generation technologies of flexible self-powered apparatus, mechanic-related sensors and actuators, large-capacity non-volatile memories, and neuromorphic networks[1–10]. However, conventional perovskite oxide systems with ample piezoelectricity and ferroelectricity (piezo/ferroelectricity) generally suffer from undesirable suppression of polar symmetry during thickness limited to a few nanometers[9,11,12]. 2D van der Waals layered materials may be a promising pathway to sidestep those challenges, and these fascinating materials have been demonstrated to have exceptional transport, magnetic and optical as well as topological properties, providing steady matter support for constructing multifunctional materials[8,13–18]. But it is unfavorable that most bulk or few layered 2D materials are centrosymmetric, such as graphene and BN of $P6/mmm$ space group[19], 2H or 1 T transition metal dichalcogenides (TMDs) with $P6_3/mmc$ or $P$-$3m1$ space group[20], and arsenene/antimonene/bismuthene ($R$−$3m$ space group)[17,21–23]. Most of the currently reported 2D piezo/ferroelectric materials focus on black phosphorus analog structures (SnS[24], SnSe[25], SnTe[26] etc.), single-layer or 1 T´ phase TMDs[27,28], CuInP$_2$S$_6$[29,30], and In$_2$Se$_3$ group[31–34], which are nothing more than a tip of the iceberg of 2D materials[35]. Accordingly, hole defining[36], twist moiré bilayer[37,38], heterostructure[39], phase transition[27,28] and Janus[40] were devised to engineer piezo/ferroelectricity in the centrosymmetric 2D layers[35].

[1]Department of Applied Physics, Hong Kong Polytechnic University, Hung Hom, Kowloon, Hong Kong, PR China. [2]Centre for Micro- and Nano-Electronics (CMNE), School of Electrical and Electronic Engineering, Nanyang Technological University, Singapore 638798, Singapore. [3]IRL 3288 CINTRA (CNRS-NTU-THALES Research Alliances), Nanyang Technological University, Singapore 637553, Singapore. ✉e-mail: apsplau@polyu.edu.hk

Recently, some unexpected piezo/ferroelectric behaviors were also discovered in several centrosymmetric 2D materials, which were explained and ascribed to crystal vacancy defects[41–43]. However, mechanistic investigations at the atomic level are absent, as well as seldom control experiments were performed to explore the regulation of piezo/ferroelectricity and the understanding of extensibility remains unclear.

Nonstoichiometric chemical compounds have been known for several decades, as a type of solid compound that deviates from the law of definite ratio[44,45]. These nonstoichiometric materials are essential, as they can generate some fantastic physical and chemical phenomena while developing technological applications[46,47]. The intuitive component engineering of 2D TMDs materials would be more readily achievable in a wide range due to their ultrathin feature and variable chemical valence of transition metal (Fig. 1)[48–51]. Moreover, unlike covalent materials, the exclusive van der Waals gap of 2D materials often generates an exceptional pathway for forming

nonstoichiometric defects, potentially bringing new insights into modifying 2D materials[48,52] (Fig. 1b, c). Considering the pervasive nonstoichiometry in TMDs and the remarkable properties of 2D TMDs, the realization of piezo/ferroelectricity in these materials is exceedingly desirable for achieving state-of-the-art multifunctional and multiferroic materials.

In this work, we artificially introduce asymmetric centers into 2D centrosymmetric TMDs through nonstoichiometric engineering executed by chalcogen vapor pressure tuning, including several representative instances of $Fe_{1+\alpha}Te_2$, $Co_{1+\beta}S_2$, $Mn_{1+\gamma}Se_2$, $Ni_{1+\delta}Se_2$, $V_{1+\varepsilon}Se_2$, $Cu_{1+\zeta}S_2$, $Cr_{1+\sigma}Se_2$. Those nonstoichiometric composites demonstrate multiple metal chemical valence states stemming from non-uniform nonstoichiometric defects. The resulting nonstoichiometric 2D $Cr_{1+\sigma}Se_2$ and $Ni_{1+\delta}Se_2$ show an obvious piezoelectric response with $d_{33}$ of 0.65 pm/V and 6.78 pm/V, respectively. $Fe_{1+\alpha}Te_2$ and $Cu_{1+\zeta}S_2$ demonstrate switchable spontaneous polarization along in-plane (IP) and out-of-plane (OOP) directions with superior ambient air stability

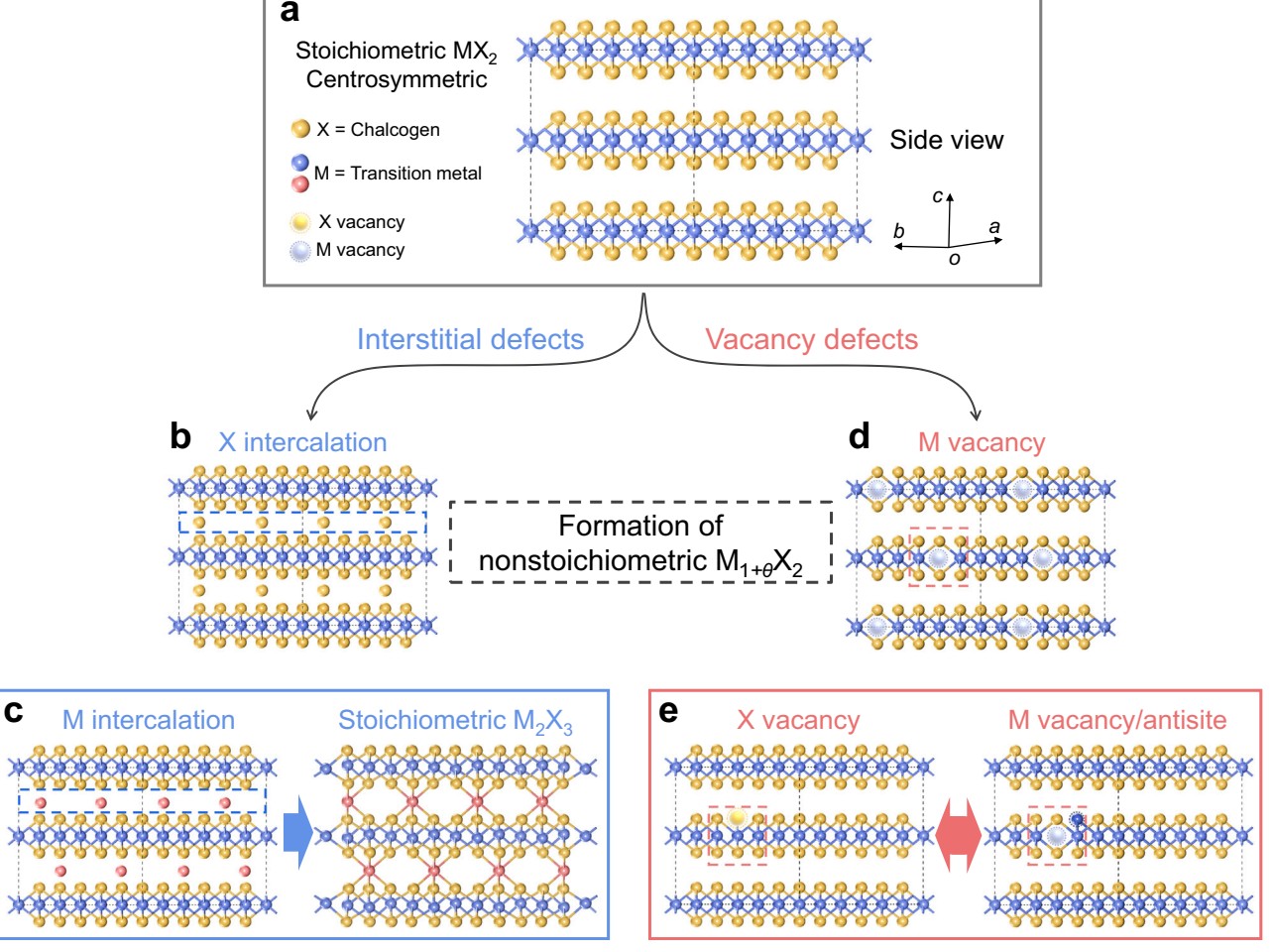

**Fig. 1 | Schematic illustration of the pathway for the formation of non-stoichiometric $M_{1+\theta}X_2$ from stoichiometric $MX_2$. a** Pristine atomic structure of three layers stoichiometric $MX_2$ with 1 T phase. This structure is layered and centrosymmetric without piezo/ferroelectricity. The yellow ball represents chalcogen atoms, while transition metal atom balls labeled in blue and red indicate primary and extra-introduced metal atoms, respectively. The metal and chalcogen atom vacancies are indicated by a ball surrounded by a dotted circle line with light yellow and light blue colors, respectively. **b** Chalcogen atoms intercalated $MX_2$ atomic structure. This structure is unstable as lack of charge compensation to chalcogen atoms. **c** Atomic structure illustration of transition metal atom intercalated $MX_2$ (left) and stoichiometric $Cr_2Se_3$ compound (right). Due to the formation of covalent bonds between transition metal atoms and chalcogen atoms, this pathway is

thermodynamically predisposed to occur. Stoichiometric $Cr_2Se_3$ is stable and centrosymmetric with a quasi-layered structure, in which sandwich layers are linked by Cr atoms. Both interlayer and intralayer Cr atoms are octahedrally coordinated by Se atoms. **d** Typical case of several random metal vacancies in layered $MX_2$. Note that a metal atom symbol in the atomic structure is overlapped by many metal atoms in the side view. The atoms are removed at this site to clearly present vacancy defects, rather than all the atoms below this site are missing. **e** A chalcogen vacancy in $MX_2$ (left) and then transfer into a metal vacancy/antisite (right). The red dash box is for eye guidance. This is just a simple possible transformation of chalcogenide vacancies. The real situation may be more complex to involve long-range atoms.

up to 6 months and a Curie temperature above 523 K. We determine that the piezoelectric coefficients are tailored through thickness and metal interstitial ion defects expressed by $\sigma/\delta$ value. Unlike commonly reported intralayer metal/chalcogen vacancies or line defects at the grain boundary, mechanical stress-induced dipole moment and switchable polarization are ascribed to interlayer metal atom displacement and van der Waals layer sliding accelerated by inhomogeneous interlayer native metal intercalation. Moreover, the nonstoichiometric 2D $Cr_{1+\sigma}Se_2$ materials demonstrate typical flexible nanogenerator behavior. The spontaneously polarized $Fe_{1+\alpha}Te_2$ nanoflakes display room-temperature magnetism, along with regulable net polarization and conductance, and exhibit distinctive multilevel memristor characteristics, holding a great potential application in non-volatile memristor and artificial intelligence apparatus.

## Results

### Formation pathways of nonstoichiometric 2D TMDs

Generally, four types of defects are formed in nonstoichiometric compounds, including anion vacancy/interstitialcy and cation vacancy/interstitialcy. For layered TMDs materials, the intralayer atoms are covalently bonded to form a sandwiched structure, and then those layers are gathered by weak van der Waals force (such as 1 T phase $MX_2$ three layers in Fig. 1a, where M is transition metal atoms and X represents chalcogen atoms). Therefore, due to relatively low energy requirements, interstitial metal or chalcogen atoms mostly occur in van der Waals gaps without breaking covalent bonds (Fig. 1b,c). However, since two sides of the sandwich layer are terminated by chalcogen atoms, the intercalation of chalcogen atoms is unstable due to the lack of electron compensation (Fig. 1b). On the contrary, metal intercalation will form covalent bonds with chalcogen atoms, releasing energy with high thermodynamic preference. A typical example is stoichiometric $Cr_2Se_3$, a stable covalent compound formed by increasing metal content to achieve a chemical ratio of Cr to Se of 2:3 (Fig. 1c). The intercalated metal atoms share part of electrons with chalcogen atoms to maintain electrical neutrality, forming multiple metal valence states. Analyzing the chemical environment of the metal atoms will help uncover this type of interstitial defect.

Vacancy defects are comparatively simple, including metal and chalcogen vacancies (Fig. 1d, e). Chalcogen vacancies compete with metal intercalations in a metal-rich/chalcogen-poor environment, while metal vacancies compete with chalcogen intercalations in a chalcogen-rich environment. Metal vacancies are the most common case in the chalcogen-abundant environment due to the unstable structure after chalcogen atom intercalation (Fig. 1d). For $MoSe_2$ and $WSe_2$ samples prepared by chemical vapor transport (CVT), chalcogen vacancies are much rarer with respect to metal vacancies or antisites, even if chalcogen vacancies have a lower formation energy[53]. This anomalous phenomenon suggests a potential transition between chalcogen vacancies and metal antisites (Fig. 1e). According to defect theory in bulk solid-state compounds, defect concentrations in nonstoichiometric 2D TMDs can be speculated as a function of chalcogen vapor pressure (Supplementary Note 1)[54]. It is found that concentrations of the metal interstitial defect and chalcogen vacancy defect are inversely proportional to the chalcogen vapor pressure, while metal vacancy defect concentration is directly proportional to the chalcogen vapor pressure. Regardless of some random factors, regulating the type and concentration of defects in the 2D TMDs layers is very reasonable by tuning the chalcogen vapor pressure. In particular, the valence states of interstitial metal atoms can be tuned over a wide range, suggesting diverse coordination types and a high possibility of metal atoms becoming embedded in 2D TMDs layers.

### Preparation and piezoelectricity of nonstoichiometric 2D TMDs

Guided by the calculations and predictions, seven representative transition metals (V, Cr, Mn, Fe, Co, Ni, and Cu, as marked as deep blue

in Fig. 2a) were selected to be prepared as nonstoichiometric 2D metal chalcogenides via tuning chalcogen pressure. The summarized space groups of those compounds with specific stoichiometric ratios are mostly centrosymmetry, which is not expected to have piezoelectricity or ferroelectricity (Supplementary Table 1). Although there are several reports on piezo/ferroelectricity in centrosymmetric 2D materials, such as CdS, $\alpha$-$Ga_2Se_3$ and $SnS_2$ (as marked in deep green in Fig. 2a)[41–43], the relationship between nonstoichiometric ratios and piezoelectric coefficients as well as the extendibility of piezo/ferroelectricity in other 2D materials are still obscure. All of the synthesized seven kinds of nonstoichiometric 2D TMDs, including $Fe_{1+\alpha}Te_2$, $Co_{1+\beta}S_2$, $Mn_{1+\gamma}Se_2$, $Ni_{1+\delta}Se_2$, $V_{1+\varepsilon}Se_2$, $Cu_{1+\zeta}S_2$, $Cr_{1+\sigma}Se_2$ ($\alpha$, $\beta$, $\gamma$, $\delta$, $\varepsilon$, $\zeta$ and $\sigma$ symbols can be positive or negative, denoting vacant or intercalated metal atoms, respectively), exhibit regular shapes of single-crystal and a reproducible collection of multiple units (Fig. 2b–g, Fig. 3a and Supplementary Fig. 1). The OOP amplitude and phase signals are strengthened with the increased drive alternating current (AC) voltage, indicating typical inverse piezoelectricity effects of those nonstoichiometric 2D TMDs (Figs. 2b–g and 3a). However, some amplitude images exhibit a non-uniform distribution of signals, which may originate from the localized nonstoichiometricity of metal atom intercalation and uneven intralayer defects. Besides, the ferroelectric domain switching and rough surface topography may also contribute to the inhomogeneous amplitude signals. Corresponding Raman and X-ray photoelectron spectroscopy (XPS) spectra confirm the successful preparation of TMDs compounds (Supplementary Fig. 2). Significantly, $2p_{3/2}$ orbits of transition metals in those nonstoichiometric 2D TMDs can be fitted by several peaks (middle panels in Supplementary Fig. 2a–g), which is typically distinct from metal oxide peaks of which binding energy is usually several eV higher than that of metal-chalcogen peaks. The splitting of metal-chalcogen XPS peaks and slight shifts in binding energy indicate different types of coordination environments and valence states of transition metal atoms, confirming the nonstoichiometric composition of the synthesized 2D TMDs.

However, confirmation of the detected signal due to intrinsic piezo/ferroelectricity is decisive to rule out spurious piezo/ferroelectric response arising from carrier injection, flexoelectricity, morphology-induced friction and electrochemical reactions promoted by the water meniscus effect[55]. Thus, to confirm piezo/ferroelectric is essentially derived from voltage-dependent amplitude variation, second harmonic generation (SHG), resonance amplification curves and top metal electrode deposition were performed on representative nonstoichiometric 2D $Cr_{1+\sigma}Se_2$ nanoflakes. The apparent second harmonic generation (SHG) mapping signals suggest the broken inversion symmetry in nonstoichiometric 2D $Cr_{1+\sigma}Se_2$ nanoflakes (Supplementary Fig. 1c). The IP and OOP amplitude intensity versus frequency at different AC voltages were also performed by piezoresponse force microscopy (PFM) on nonstoichiometric 2D $Cr_{1+\sigma}Se_2$ nanoflakes (Supplementary Fig. 3). The peak frequencies at about 728 kHz and 424 kHz conform to IP and OOP resonance peaks, respectively. Compared with the IP mode, the OOP mode shows more noticeable amplitude signal variation related to the drive voltages, demonstrating inherent piezoelectricity and expected linear behavior (Supplementary Fig. 3c, d). The enhanced OOP piezoresponse compared with IP piezoresponse may be attributed to the thinner feature in the vertical direction, resulting in the direct and rapid feedback of piezoelectric amplitude rather than dissipated with lattice relaxation. Beyond, a 20 nm thick Au film was deposited on a 3.6 nm nonstoichiometric 2D $Cr_{1+\sigma}Se_2$ nanoflake on mica substrate to exclude potential artifacts triggered by surface charging, flexoelectricity or air-environmental electrochemical reaction (Supplementary Fig. 4). No noticeable difference was observed before and after Au deposition, indicating real piezoelectricity in nonstoichiometric 2D TMDs. Especially, the amplitude of the sample changes with driving voltage, while substrate amplitude is almost consistent, confirming that the substrate supplies

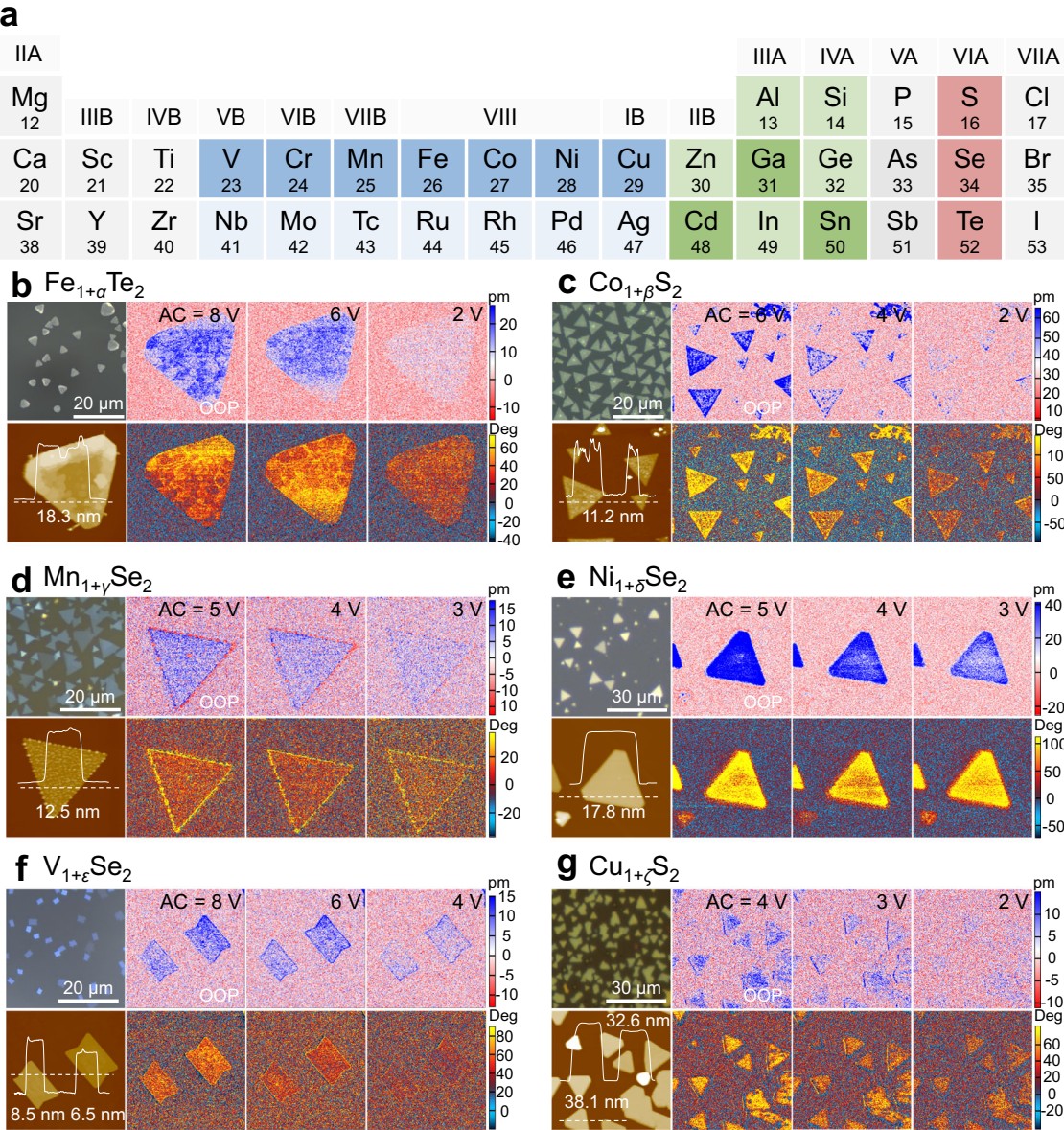

**Fig. 2 | A library summary of OOP piezo/ferroelectricity in nonstoichiometric 2D metal chalcogenides. a** Periodic table illustration of metal elements and chalcogen elements for the formation of nonstoichiometric two-dimensional (2D) metal chalcogenides. Deep green indicates already reported piezo/ferroelectricity in their metal chalcogenides compounds, while light green indicates congeners with intense green. The dark blue suggests nonstoichiometric 2D transition metal dichalcogenides (TMDs) that are experimentally realized and identified as piezo/ferroelectricity in this work. Light blue color indicates congeners with dark blue.

The orange symbolizes chalcogens for preparing 2D metal chalcogenides and nonstoichiometric 2D TMDs. **b–g** Optical images (upper left), atomic force microscopy (AFM) images and thickness profile (lower left), out-of-plane (OOP) amplitude images driven by different alternating current (AC) voltages (upper right) and corresponding phase images (lower right) of nonstoichiometric 2D $Fe_{1+\alpha}Te_2$ (**b**), $Co_{1+\beta}S_2$ (**c**), $Mn_{1+\gamma}Se_2$ (**d**), $Ni_{1+\delta}Se_2$ (**e**), $V_{1+\varepsilon}Se_2$ (**f**), $Cu_{1+\zeta}S_2$ (**g**). Each nonstoichiometric 2D TMDs exhibits a voltage-dependent amplitude, indicating a pronounced piezoelectric effect.

no contribution to the total piezoelectric response of the nanoflakes and that the piezoelectricity originates from the inverse piezoelectric effect of nanoflakes.

**Tuning piezoelectric coefficient in nonstoichiometric 2D TMDs**

Although the intrinsic amplitude at a single drive AC voltage can directly calculate the effective piezoelectric coefficient, local and single test fluctuations may affect the accuracy of the value of the real piezoelectric coefficient. Thus, by fitting multiple points of intrinsic piezoelectric amplitude of the sample to the drive AC voltages, the effective $d_{33}$ of a 5.6 nm thick nonstoichiometric 2D $Cr_{1+\sigma}Se_2$ nanoflake and a 47.8 nm thick nonstoichiometric 2D $Ni_{1+\delta}Se_2$ nanoflake were counted to be $0.65 \pm 0.12$ pm/V and

$6.78 \pm 0.6$ pm/V, respectively (Fig. 3b, more details are elaborated in Supplementary Note 2 and Supplementary Figs. 5–11). The effective $d_{33}$ of nonstoichiometric 2D $Cr_{1+\sigma}Se_2$ nanoflake is almost the same as some reported 2D piezoelectrics (Supplementary Table 2), such as intrinsic piezoelectric 3R-MoS$_2$ (0.9 pm/V)[56] and $\alpha$-Tellurium film (1 pm/V)[57] as well as acquired/engineered piezoelectrics of MoO$_2$ (0.56 pm/V)[58] and doped graphene (1 pm/V)[19]. The advantages of the engineered piezoelectric properties of this material are even more substantial if thickness concerns are involved (Supplementary Table 2). Especially, the thick nonstoichiometric 2D $Ni_{1+\delta}Se_2$ nanoflake shows superior piezoelectric performance, even comparable to well-known intrinsic 2D $\alpha$-In$_2$Se$_3$ piezoelectrics (Supplementary Table 2)[59].

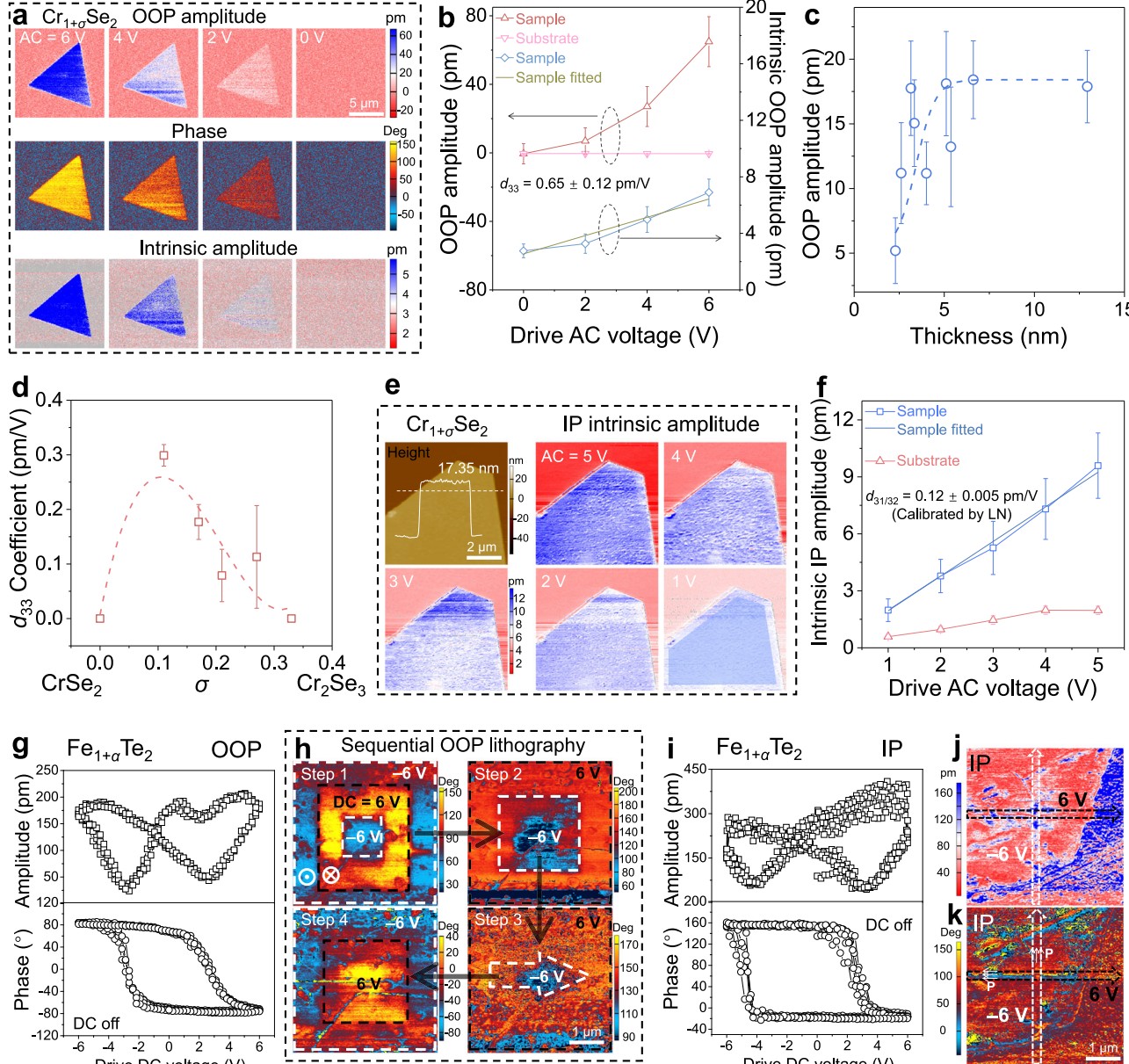

**Fig. 3 | Tuning piezoelectricity of nonstoichiometric $Cr_{1+\sigma}Se_2$ nanoflakes and emerging ferroelectricity in nonstoichiometric $Fe_{1+\alpha}Te_2$ nanoflakes. a** OOP amplitude images (up), phase images (middle) and intrinsic amplitude images (bottom) of a nonstoichiometric $Cr_{1+\sigma}Se_2$ nanoflake under different drive voltages. **b** Corresponding OOP amplitude evolution curve with drive AC voltage. Orange, pink, blue and prasinous lines are assigned to the resonance-amplified OOP amplitude of the sample, resonance-amplified OOP amplitude of the substrate, original intrinsic OOP amplitude of the sample and linearly fitted intrinsic OOP amplitude of the sample as a function of drive AC voltage, respectively. **c** Evolution of resonance-amplified OOP amplitudes of nonstoichiometric $Cr_{1+\sigma}Se_2$ nanoflakes with different thicknesses. **d** Effective $d_{33}$ of nonstoichiometric $Cr_{1+\sigma}Se_2$ nanoflakes with different values of $\sigma$. **e** Height image and different AC voltage-driven intrinsic amplitude images of a nonstoichiometric $Cr_{1+\sigma}Se_2$ nanoflake. **f** Corresponding

intrinsic IP amplitude in **e** as a function of the drive AC voltage of substrate and nanoflake sample. Red, blue and dark blue lines represent the IP intrinsic amplitude of the substrate and IP intrinsic amplitude of sample before and after fitted, respectively. **g** Local OOP ferroelectric switching spectra under direct current (DC) off state of a nonstoichiometric $Fe_{1+\alpha}Te_2$ nanoflake. **h** Phase images of a nonstoichiometric $Fe_{1+\alpha}Te_2$ nanoflake after sequential lithography by using different patterns. The black and white dotted line frames indicate +6 V and -6V lithography area, respectively. **i** Local IP ferroelectric hysteresis loops under DC off state of a nonstoichiometric $Fe_{1+\alpha}Te_2$ nanoflake. **j**–**k** IP amplitude (**j**) and IP phase (**k**) images of a nonstoichiometric $Fe_{1+\alpha}Te_2$ nanoflake after electrical field writing along two perpendicular lines. The black line and white line indicate first +6 V and then –6V lithography. All the error bars indicate standard deviation.

The piezoelectric coefficient of 2D materials is also normally associated with thickness. Thus, a large-area PFM scanning of non-stoichiometric 2D $Cr_{1+\sigma}Se_2$ nanoflakes covering different thicknesses was performed to evaluate the variation tendency of amplitude with thickness (Supplementary Fig. 6). PFM characterizations of nanoflakes with varying thicknesses in the same window can screen external disturbing and ensure a credible comparison. The obtained relationship curve shows a behavior of increasing first and then saturating, identical to the discovery of intrinsic piezoelectric $\alpha$-$In_2Se_3$ (Fig. 3c)[59]. Besides, SHG mapping images also show enhanced intensity on darker (thicker) nanoflakes (Supplementary Fig. 1). The initial increase of piezoelectric amplitude with thickness may be attributed to the reduction of substrate induction and decoupling between nanoflake and substrate. In addition to thickness, the chemical composition may primarily

determine the piezoelectric coefficient, especially for nonstoichiometric 2D TMDs. The chemical composition tuned $Cr_{1+o}Se_2$ and $Ni_{1+\delta}Se_2$ nanoflakes show a noticeable first decrease and then increase of effective $d_{33}$ with the increase of $\sigma/\delta$ value (Fig. 3d, Supplementary Figs. 7–11, Supplementary Note 3). Therefore, the piezoelectric coefficient is more sensitive to the initial metal intercalation/vacancy. A small number of defects can rapidly enhance the piezoelectric coefficient, highlighting the significance of the nonstoichiometric ratio in engineering piezoelectricity in 2D materials.

Consistent with the IP resonance amplification curves, IP PFM mapping images also show some degree of drive voltage-dependent IP amplitudes (Supplementary Fig. 12, Supplementary Note 2). The high-resolution IP PFM characterizations of $Cr_{1+o}Se_2$ and $Ni_{1+\delta}Se_2$ nanoflakes can more distinctly reveal the evolution of IP intrinsic amplitude relevant to drive AC voltage (Fig. 3e, f and Supplementary Fig. 13). To accurately describe the IP piezoelectric coefficient, it is crucial to detect the angle-resolved IP piezoelectric response to specify the contribution of $d_{31}$ and $d_{32}$. Thus, IP PFM characterizations were performed on three $Ni_{1+\delta}Se_2$ nanoflakes with different alignment directions (Supplementary Fig. 14). As the six-fold symmetry of the hexagonal crystals, the nearly identical IP piezoelectric coefficients of the three nanoflakes are sufficient to identify the isotropy of the IP piezoelectric response of the nonstoichiometric 2D nanoflakes (Supplementary Fig. 14). Thus, after calibrated by $z$-cut lithium niobate (LN) crystals, effective $d_{31/32}$ of $Cr_{1+o}Se_2$ and $Ni_{1+\delta}Se_2$ nanoflake were calculated to be $0.12 \pm 0.005$ pm/V and $0.16 \pm 0.017$ pm/V, respectively (Fig. 3e, f and Supplementary Fig. 13). The IP piezoelectric coefficient is not very impressive, but is sufficient to indicate essential IP piezoelectric response in nonstoichiometric nanoflakes and application in nanogenerators.

## Ferroelectricity in nonstoichiometric 2D TMDs

PFM amplitude and phase images of several as-prepared nonstoichiometric 2D TMDs, such as $Fe_{1+\alpha}Te_2$ and $Cu_{1+\zeta}S_2$, show an inhomogeneous distribution of intensity (Fig. 2b, g). The magnified amplitude and phase images collected in IP and OOP modes demonstrate distinct and sharp domains, suggesting the potential ferroelectricity of $Fe_{1+\alpha}Te_2$ (Supplementary Fig. 15, Supplementary Note 4). Then, local switching spectroscopy was conducted on 2D $Fe_{1+\alpha}Te_2$ nanoflakes to verify ferroelectric polarization. The PFM amplitude and phase collected in OOP and IP configuration of nonstoichiometric 2D $Fe_{1+\alpha}Te_2$ as a function of the drive direct current (DC) voltage exhibit typical butterfly shapes and hysteresis loops (Fig. 3g, i), confirming its ferroelectric features. The shift of phase difference from 180° may be attributed to local electrostatic and charging effects.

To further confirm the exotic ferroelectric behavior, the OOP polarization lithography was undertaken by applying ±6 V DC voltage on $Fe_{1+\alpha}Te_2$ (white and black dotted lines in Fig. 3h). Subsequent PFM scanning presents two contrast areas in the phase image, suggesting the large-area switchable spontaneous polarization of nonstoichiometric 2D $Fe_{1+\alpha}Te_2$ (up left in Fig. 3h). After sequential electrical field writing of different patterns and then conducting PFM imaging, the ferroelectric domains of $Fe_{1+\alpha}Te_2$ nanoflakes correspondingly switch, showing distinct signal differences even after 4 times of lithography (Fig. 3h). Besides, the switching of IP spontaneous polaritons of nonstoichiometric 2D $Fe_{1+\alpha}Te_2$ nanoflakes was also verified by sequentially scanning a black line with 6 V and a white line with –6 V voltage along the direction of the arrow in Fig. 3j, k. The first lithography with a positive voltage (black line) shows obvious manipulation of the polarization direction (Supplementary Note 4, Fig. 3k), and subsequent negative voltage line, in turn, changes the direction of ferroelectric polarization (as displayed in the intersection in Fig. 3k). The multiple lithography of ferroelectric domains suggests rewritable and robust polarization of nonstoichiometric 2D $Fe_{1+\alpha}Te_2$ nanoflakes. Corresponding OOP electric-hysteresis behavior and polarization

direction reversal can also be observed and implemented in $Cu_{1+\zeta}S_2$ nanoflakes (Supplementary Fig. 16), confirming the extendable ferroelectricity through nonstoichiometric engineering. Especially, morphology-independent phase and amplitude images of nonstoichiometric nanoflakes exclude the pseudo-ferroelectricity originating from surface topography or oxidation/damage (Supplementary Figs. 15, 16c, 17 and 18b). The incomplete 180° flipping of the ferroelectric domains during lithography may be due to short dwelling time during electrical field writing or tilted polarization direction of the lowest energy instead of absolutely in-plane or out-of-plane[28].

Remarkably, the nonstoichiometric 2D $Fe_{1+\alpha}Te_2$ nanoflakes used for PFM characterization in Fig. 3g–k have been stored in an air environment for 6 months, demonstrating remarkable ferroelectric response compared with freshly prepared nanoflakes and suggesting superior stability of the ferroelectric phase (Supplementary Fig. 18). The high stability of nonstoichiometric 2D TMDs nanoflakes can also be identified by the temperature-dependent Raman spectra of both $Fe_{1+\alpha}Te_2$ and $Cu_{1+\zeta}S_2$ nanoflakes (Supplementary Fig. 19), indicating that both Curie temperatures are above 523 K. The excellent stability may be attributed to the covalent bonding of the intercalated metal ions with the van der Waals layer, thereby enhancing the chemical coordination saturation of the chalcogen atoms and inhibiting the intrusion of oxygen and the degradation of the crystal structure. By combining the characterizations of high-resolution ferroelectric domains, PFM hysteresis loops and multiple artificial lithography, it can be determined that the spontaneous polarization mostly originated from the intrinsic ferroelectricity of the nonstoichiometric nanoflakes rather than external spurious response[55]. The spontaneous polarization of nonstoichiometric 2D nanoflakes reveals that nonstoichiometric engineering is an effective approach for accessing the ferroelectricity of 2D materials.

## Atomic-scale characterizations of nonstoichiometric 2D materials

To further investigate the atomic structure of nonstoichiometric 2D materials, the nonstoichiometric $Cr_{1+o}Se_2$ nanoflakes were selected as an example and transferred onto Cu grids for high-resolution transmission electron microscopy (HRTEM) characterizations. Low-magnification TEM image and X-ray energy dispersive spectroscopy (EDS) elemental mapping results of $Cr_{1+o}Se_2$ nanoflakes demonstrate integrated triangle shape and uniform signal distribution with almost indistinguishable oxygen signals related to carbon grids, confirming that the transfer process is reliable without degradation and the nanoflakes are composed of Cr and Se (Supplementary Fig. 20a, b). The top-view TEM image along the [001] zone axis presents regular interference patterns but with uneven intensity distribution, as marked by blue and red dotted circles for dark and light sites (Fig. 4a). However, stimulated HRTEM images of $Cr_2Se_3$ show homogeneous density (Fig. 4b). The distinguished signals can also be found in selected area electron diffraction (SAED) patterns of an experimental nonstoichiometric $Cr_{1+o}Se_2$ nanoflake and simulated $Cr_2Se_3$ (Supplementary Fig. 20c and Fig. 4c). Similar uneven diffraction spots and interference sites can also be observed in nonstoichiometric $Ni_{1+\delta}Se_2$ nanoflake (Supplementary Fig. 21). The differences in HRTEM image contrast and SAED patterns imply that the nonstoichiometric $Cr_{1+o}Se_2$ and $Ni_{1+\delta}Se_2$ nanoflakes may contain numerous vacancy defects or extra metal atoms intercalations compared with the stoichiometric compounds, which may lead to spatial inversion symmetry breaking and apparent IP and OOP piezoelectricity. Besides, the regular single-point array and in-plane six-fold symmetry in experimental SAED patterns of nonstoichiometric nanoflakes confirm the well-maintained single-crystal characteristics after nonstoichiometric engineering.

High-resolution scanning TEM (HRSTEM) characterizations in high-angle annular dark-field (HAADF) and integrated differential phase contrast (iDPC) imaging were then performed to reveal the

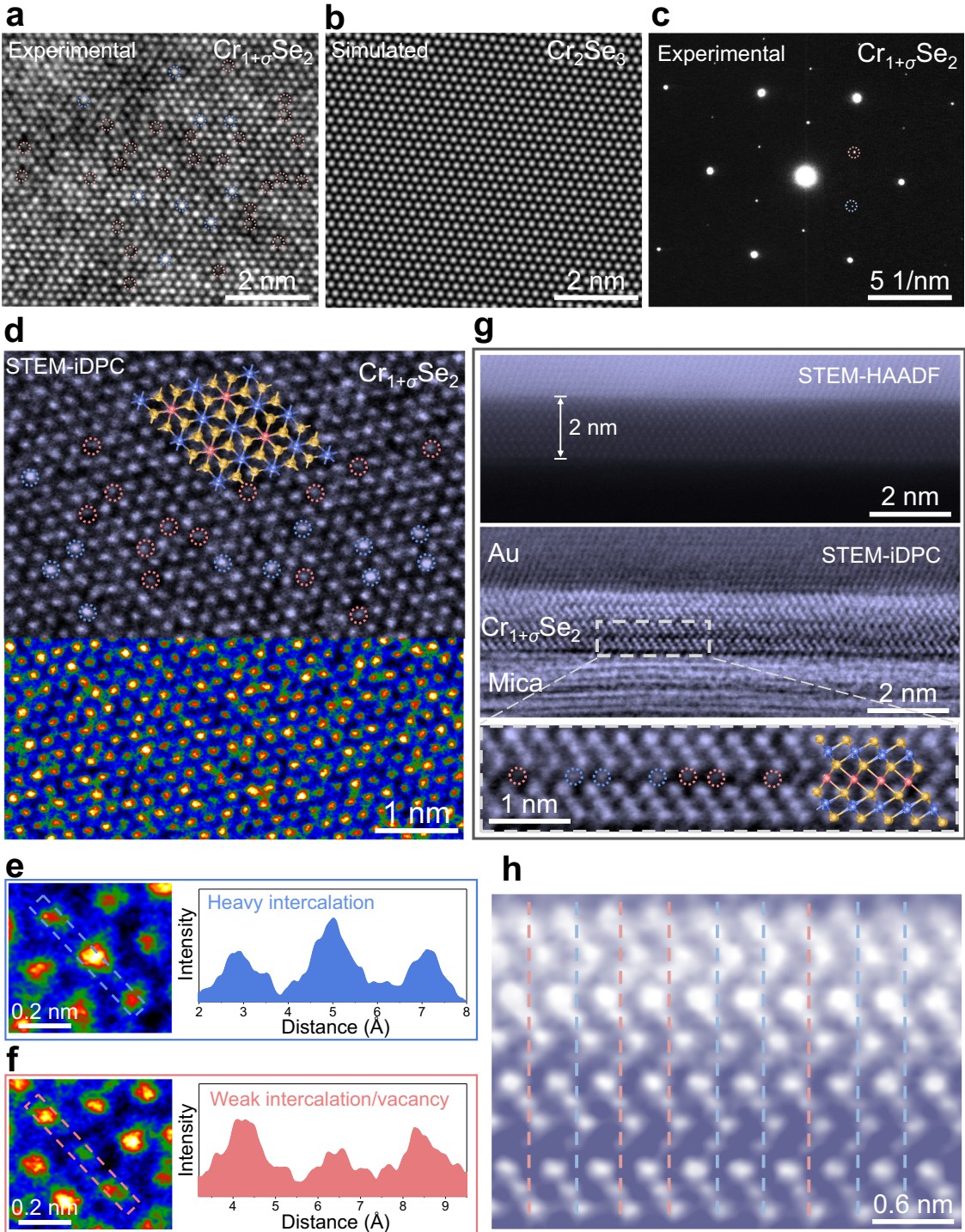

**Fig. 4 | Atomic-scale characterizations of the nonstoichiometric 2D $Cr_{1+\sigma}Se_2$ nanoflake. a–c** Experimental high-resolution transmission electron microscopy (HRTEM) image of a triangular nonstoichiometric 2D $Cr_{1+\sigma}Se_2$ nanoflake (**a**), stimulated HRTEM image of standard stoichiometric $Cr_2Se_3$ (**b**) and experimental selected area electron diffraction (SAED) patterns of a triangular nonstoichiometric 2D $Cr_{1+\sigma}Se_2$ nanoflake (**c**). Blue dotted circles and red dotted circles point out interference sites with heavy intercalation/weak vacancy and weak intercalation/heavy vacancy defects, respectively. **d** Top-view scanning transmission electron microscopy (STEM)-integrated differential phase contrast (iDPC) image of a triangular nonstoichiometric 2D $Cr_{1+\sigma}Se_2$ nanoflake. Blue dotted circles and red dotted circles demonstrate heavy and light Cr atoms intercalation, respectively. **e–f** Magnified STEM-iDPC images and corresponding intensity line profiles of a heavy intercalation defect (**e**) and a weak intercalation/vacancy defect (**f**). **g–h** Cross-sectional STEM high-angle annular dark-field (STEM-HAADF) image and corresponding STEM-iDPC image (**g**) and average background subtraction filter (ABSF) filtered cross-sectional STEM-iDPC image with high magnification (**h**) of a nonstoichiometric 2D $Cr_{1+\sigma}Se_2$ nanoflake. Blue dotted circles and red dotted circles demonstrate heavy and weak Cr atoms intercalation, respectively. Blue and red dotted lines are for eye guidance of van der Waals layer sliding. Blue and red dotted lines indicate the displacement and non-displacement of interacted metal atoms, respectively.

accurate spatial distribution of these defects in nonstoichiometric $Cr_{1+\sigma}Se_2$ nanoflakes at the atomic-scale. According to the point-by-point scanning mechanism in STEM imaging, the Cr or Se atoms can be directly marked in STEM-iDPC and STEM-HAADF images (Supplementary Fig. 20d and Fig. 4d). Parallel to the HRTEM image, the top-view STEM-iDPC image collected along the [001] axis also shows randomly distributed light and dark atomic sites in this Z-contrast image, indicating heavy intercalation defects or weak intercalation/vacancy defects, respectively (Fig. 4d–f). In a wide field view, STEM images also demonstrate the randomly distributed interstitial defects over a large area in the nonstoichiometric 2D $Cr_{1+\sigma}Se_2$ nanoflakes, suggesting the large-scale and generally existed intercalation rather than in a small localized region (Supplementary Fig. 22).

Since Cr and Se atoms are vertically stacked along [001] direction, it is difficult to distinguish the metal and chalcogen atoms from a planar view. Therefore, cross-sectional STEM characterizations of a 2 nm thick nonstoichiometric $Cr_{1+\sigma}Se_2$ nanoflake were subsequently performed in detail, revealing the AA stacking sequence of the $CrSe_2$ bone layer and the interlayer non-uniform intercalation of Cr atoms (Fig. 4g). The interlayer metal intercalation is well consistent with the metal-rich EDS composition statistics (Supplementary Fig. 7). No apparent defects are observed in the $CrSe_2$ bone layers, confirming a relatively high energy barrier for intralayer atom intercalation (Fig. 4g). However, the sliding of the van der Waals layers can be observed by vertical alignment of the intralayer atoms in a magnified and filtered cross-sectional STEM-iDPC image (Fig. 4h), demonstrating an increased sliding distance from the top layer to the bottom layer. Besides, interlayer metal atoms also display specific displacements along the $z$ direction (blue lines in Fig. 4h). TEM and EDS investigations were also implemented on another nonstoichiometric 2D $Fe_{1+\alpha}Te_2$ nanoflake with ferroelectricity. Similar to experimental observations in $Cr_{1+\sigma}Se_2$, the uneven diffraction spots in SAED patterns, inhomogeneous interference stripes in the HRTEM image, uniform elemental distribution and extra metal contents in EDS mapping were all observed in this nonstoichiometric 2D $Fe_{1+\alpha}Te_2$ nanoflakes (Supplementary Figs. 23 and 24).

Those detailed electron microscopic analyses reliably uncover that the formation mechanism of nonstoichiometric 2D compounds is mostly attributed to atomic-level interlayer metal intercalations between sandwich $MX_2$ layers and potential slight intralayer defects. The partially intercalated metal atoms covalently bond with chalcogen atoms in the bone $MX_2$ layer, forming completely and partially coordinated chalcogen atoms. The heterogeneous van der Waals force and covalent bonding interaction may cause fluctuations in the van der Waals layers, which are responsible for local interlayer sliding and metal atom displacement. The partial intercalation of metal atoms breaks the center equilibrium of positive and negative units, producing net electric dipole moments while suffering from external mechanical stress. Besides, the sliding in the $MX_2$ layer can create an inequivalent between adjacent layers, producing the interlayer spontaneous alignment of electric dipoles along the OOP direction. The reduced potential energy barrier of interlayer sliding ensures the essential switching of the ferroelectric domain under an external electric field[28,60–62]. In addition to interlayer sliding, the structural distortion induced by asymmetrical metal coordination and potential intralayer defects also accelerate the occurrence of lateral electric dipole and yield stable IP polarization[63–65]. Switching of the external electrical field can control the lateral displacement of the intercalated metal ions and intralayer defects, thereby reversing the direction of IP spontaneous polarization in nonstoichiometric 2D TMDs.

## Applications of nonstoichiometric 2D materials

Two-terminal devices based on nonstoichiometric 2D materials were fabricated to study electrical properties and demonstrate these potential applications. As the ultrathin and flexible merits of 2D

materials, a large degree of stretch can be implemented on a non-stoichiometric 2D $Cr_{1+\sigma}Se_2$ nanoflake device on polyimide (PI) substrate fixed on the finger (Fig. 5a). During bending and straightening processes, short-current and open-circuit voltage responses were obtained in two opposite signs, directly confirming mechanical to electrical energy conversion of nonstoichiometric 2D $Cr_{1+\sigma}Se_2$ nanoflakes (Fig. 5b). Corresponding multiperiodic strains produce stable and constant currents and voltage output with a peak voltage of about 200 mV (Fig. 5c, d), indicating potential applications of nonstoichiometric 2D materials in piezoelectric nanogenerators and self-powered sensing/detecting apparatus[66].

Multiferroicity is also of high possibility to be realized on these 2D crystals since transition metals such as Fe, Co, Ni, Mn, V and Cr are famous for their magnetism with uncompensated spins on 3d orbit. As a verification, magnetic hysteresis loops are demonstrated in non-stoichiometric 2D $Fe_{1+\alpha}Te_2$ nanoflakes at 10 K and 300 K, indicating the simultaneous existence of room-temperature magnetism and ferroelectricity in the stoichiometrically engineered 2D TMDs (Supplementary Fig. 25). The IP and OOP ferroelectricity of the 2D $Fe_{1+\alpha}Te_2$ nanoflakes make them candidates for the ferroelectric tunnel junction (FTJ) and ferroelectric memristor applications. Firstly, the vertical electrical transport behaviors were investigated to elaborate OOP ferroelectricity via conductive force microscopy (CFM) (Fig. 5e). The high voltages (more negative or positive) reverse the polarization direction of the ferroelectric nanoflakes, leading to the two different resistances associated with different $I$-$V$ curve slopes (Fig. 5e).

The enlarging loop windows with the increase of voltage range substantially verify the OOP ferroelectricity and signify the FTJ application (Fig. 5e). However, higher voltage also potentially enhances the tunneling current and causes the loop deviation (as indicated by the $I$-$V$ curve with a sweeping range of –10 V to 10 V in Fig. 5e). IP electrical characteristics of two-terminal devices based on a 2D $Fe_{1+\alpha}Te_2$ nanoflake were also measured in different voltage sweeping ranges. The $I$-$V$ curves of a 2D $Fe_{1+\alpha}Te_2$ nanoflake demonstrate a similar current hysteresis loop with typical high and low resistance states, confirming its IP ferroelectricity (Fig. 5f and Supplementary Fig. 26a). With a decrease in maximum voltage, the difference in current between the two states is also decreased, indicating multilevel states of the ferroelectric memristor (Fig. 5f). The curve shape is symmetric in the negative and positive voltage sweeping ranges, meaning a small Schottky barrier. However, a special feature is the presence of non-zero current at the zero voltage during sweeping, suggesting a charging and discharging process. This behavior may be induced by defect charge trapping, inner net polariton capacity or metal-nanoflake interface, which were also shown in the in-plane devices of $\alpha$-$In_2Se_3$ ultrathin nanoflakes with intrinsic ferroelectricity[67]. The open circuit voltage field (OCVF) is highly tunable and dependent on the sweeping range (Supplementary Fig. 26b). Moreover, the transistor shows considerable stability in switching behavior and a relatively stable OCVF even after 109 cycles of sweeping (Fig. 5g, Supplementary Fig. 26c). The controllable and stable charging and releasing process endows peculiar characteristics for this type of material. For example, the device shows certain retention performance after poling with high and low resistance states (Fig. 5h).

The initial decrease of currents may have originated from charge consumption or polarization self-release. A suppressed current situation was also observed after the cyclic voltage sweep (Fig. 5i). With the increase of voltage sweeping cycles, the drain current decreases and the open circuit voltages increase correspondingly, which mimics the synaptic dynamics of long-term plasticity (Fig. 5i–j). These results also suggest that the electrical behavior of 2D $Fe_{1+\alpha}Te_2$ nanoflake is dually controlled by polarization and charging. The good stability and cyclic endurance indicate that nonstoichiometric 2D $Fe_{1+\alpha}Te_2$ nanoflakes are promisingly applied for multilevel memristors and synapses.

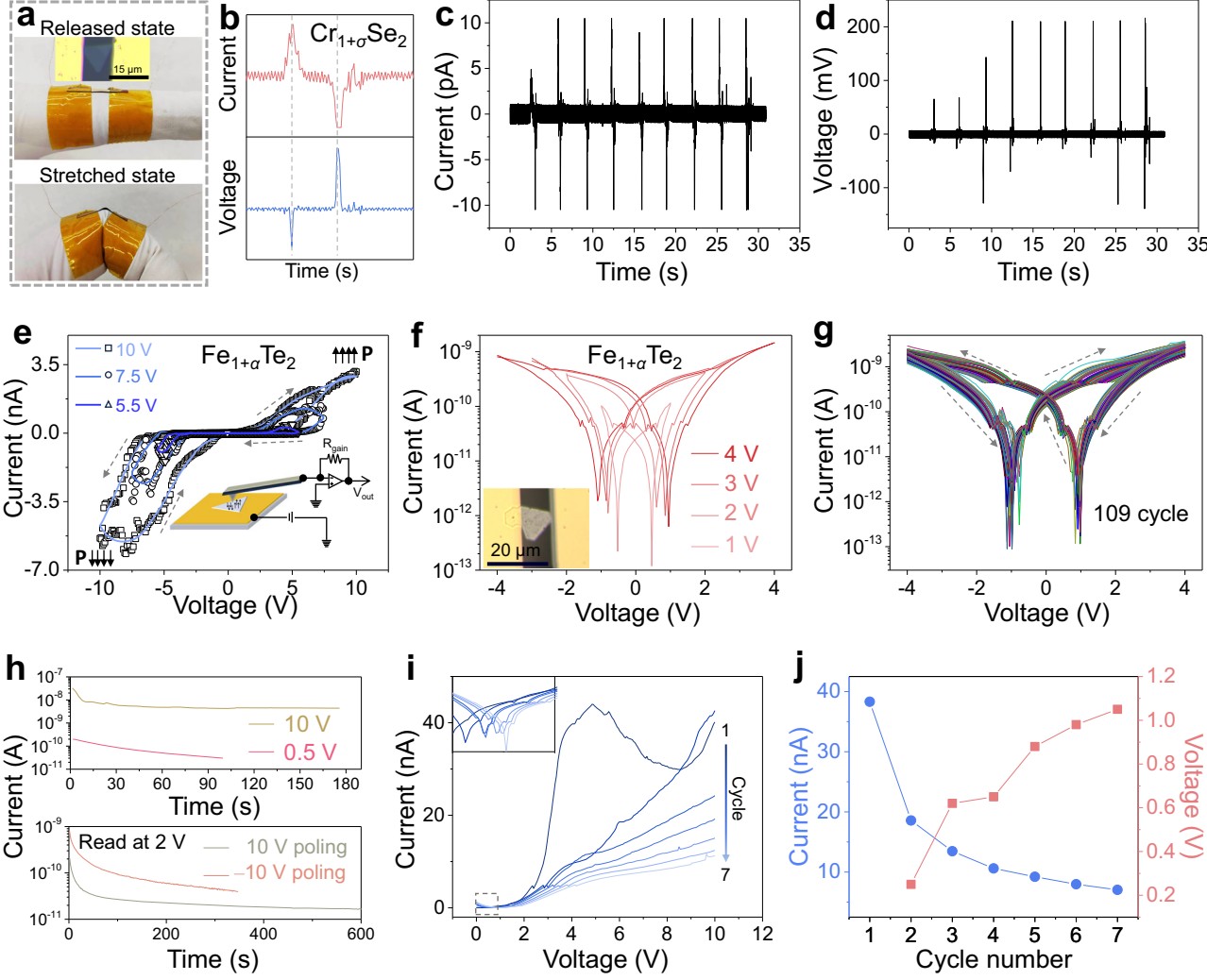

**Fig. 5 | Electrical behavior of two-terminal devices based on nonstoichiometric 2D materials. a** Digit photo images and an optical image of a two-terminal nanogenerator based on a nonstoichiometric 2D $Cr_{1+\sigma}Se_2$ nanoflake. **b** Output voltage and short circuit current of a 2D $Cr_{1+\sigma}Se_2$ nanoflake under one release-stretch-release cycle. **c–d** Output voltage and short circuit current as a function of time under several cycles of tensile strain. **e** Electrical transport behaviors of the nonstoichiometric 2D $Fe_{1+a}Te_2$ nanoflakes along OOP direction. The inset illustrates the principal scheme of conductive force microscopy (CFM) using a conductive tip and Au layer as top and bottom electrodes, respectively. The lines with colors changed from deep blue to light blue represent the fitted curves tested at the voltages ranged from 10 V, 7.5 V to 5.5 V, respectively. **f** I-V curves of a two-terminal nonstoichiometric 2D $Fe_{1+a}Te_2$ nanoflake device with different maximum sweeping

voltages. The red to light red lines indicate the I-V curves tested at the voltage of 4 V, 3 V, 2 V and 1 V, respectively. **g** I-V curves of a two-terminal nonstoichiometric 2D $Fe_{1+a}Te_2$ nanoflake device with 109 cycle sweeping. The sweeping voltage is set in a range of –4 to 4 V. **h** Drain currents retention curves read at drain voltages of 10 V (khaki line) and 0.5 V (red line) in the up panel and read at 2 V after 10 V (dark khaki line) and –10 V (orange line) bias poling (below panel). **i** I-V curves after 7 times accumulation indicated by lines with colors evolved from dark blue to light blue. The inset shows the enlarged grey dotted frame of the curves in logarithmic coordinates. Drain voltage sweeps from 0 V to 10 V in each cycle. **j** Currents (blue dots and line) and open circuit voltages (red squares and line) evolution in the 7 cycles sweeping.

## Discussion

In summary, our study demonstrates several instances of piezo/ferroelectricity engineering in 2D centrosymmetric TMDs by strategically manipulating stoichiometric ratios. This inherently expectable and reasonable approach surprisingly showcases remarkable extensibility, significantly broadening the ultrathin 2D piezo/ferroelectrics library. Moreover, distinct from the well-known nonstoichiometric defects in bulk solids, the controllable self-metal intercalation within the van der Waals gaps in these 2D materials represents a groundbreaking methodology for producing interlayer sliding and metal atoms displacement and engineering piezoelectric response and switchable aligned dipole. In addition, the covalent bonds formed by the intercalated metal atoms and chalcogen atoms also potentially enhance the phase and crystal stability of the nonstoichiometric 2D TMDs. Finally, piezo/ferroelectricity-based applications, such as nanogenerator and

ferroelectric memristor, were also successfully demonstrated on these nonstoichiometric 2D materials. This engineering paradigm can seamlessly integrate piezo/ferroelectricity with diverse 2D materials, facilitating the construction of multifunctional materials and innovative functional devices.

## Methods

### Overall summary of the sample preparation

All the nonstoichiometric 2D materials were prepared in a 1-inch double-zone tube furnace (TL1200–1200, Boyuntong Instrument Technology Co. Nanjing, China). Metal chlorides and chalcogen substance powder, mica and a mixture gas of hydrogen and argon (with an $H_2$ volume fraction of 5%) served as precursors, substrates and carrier gas, respectively. The carrier gas flow was set to 500 sccm during the 10 min purge stage and 100 sccm at the furnace ramping and material

growth duration. In the case of samples prepared in the second batch, the experimental parameters and instruments are detailed accordingly. All of the samples prepared in the second batch will be noted.

## Growth of $Fe_{1+\alpha}Te_2$

The growth of $Fe_{1+\alpha}Te_2$ used $FeCl_2$ and Te powder as precursors. Te powder (1 g, 99.99%, Macklin) was placed in a quartz boat in the center of the upstream furnace zone. Anhydrous $FeCl_2$ (20 mg, 99.5%, Aladdin) powder was spread and placed in a quartz tile in the center of the downstream furnace zone. Freshly exfoliated mica substrates were placed on the quartz tile and facing down the $FeCl_2$ powder. Subsequently, the upstream and downstream zones were heated to 475 °C and 530 °C within 30 min and then maintained for 10 min. After that, the furnace naturally cooled to room temperature.

The samples used in Supplementary Fig. 15d–f were prepared in the second batch. The experiments were prepared in a 2-inch tube furnace with a single heating zone. Te particles (6 g, 99.99%, Sigma-Aldrich) were placed in a quartz boat upstream and anhydrous $FeCl_2$ powders (4.5 mg, 98%, Sigma-Aldrich) were set in a quartz tile in the center of the tube furnace zone, respectively. Freshly exfoliated mica was used as the substrate, which faced down the $FeCl_2$ powder. The tube chamber was first purged by 100 sccm Ar and 90 sccm $H_2$ gases for 30 min, then the flow was maintained. Subsequently, the furnace zone was heated to 530 °C within 30 min and then held for 10 min. The temperature of the Te precursor was estimated to be ~475 °C during the material growth stage. After that, the furnace naturally cooled to room temperature.

## Growth of $Co_{1+\beta}S_2$

The growth of $Co_{1+\beta}S_2$ used $CoCl_2$ and S powder as precursors. S powder (150 mg, 99.99%, Macklin reagent) was placed in a quartz boat upstream outside the furnace zone. A quartz tile with $CoCl_2$ (5 mg, 98%, Sigma-Aldrich) powder was located at the center of the downstream furnace zone. Freshly exfoliated mica substrates were obliquely placed on quartz tile above the $CoCl_2$ powder. Subsequently, the $CoCl_2$ zone was ramped to 550 °C, and S powder was heated to 125 °C via a heating belt for 25 min, and then maintained for 10 min for the materials growth. After that, the furnace was naturally cooled to room temperature.

## Growth of $Mn_{1+\gamma}Se_2$

$Mn_{1+\gamma}Se_2$ growth was analogous to $Fe_{1+\alpha}Te_2$ growth procedure, while metal and chalcogen precursors were $MnCl_2$ powder (20 mg, 98%, International laboratory) and Se powder (300 mg, 99.99%, Aladdin), respectively. The temperature of the upstream and downstream zones was adjusted to 330 °C and 630 °C, respectively.

## Growth of $Ni_{1+\delta}Se_2$

The growth of $Ni_{1+\delta}Se_2$ was similar to the growth of $Fe_{1+\alpha}Te_2$. However, $NiCl_2 \cdot 6H_2O$ powder (10 mg, 98%, Tianjin Damao) and Se powder (300 mg, 99.99%, Aladdin) were employed as precursors. The growth temperature of the upstream zone and the downstream zone was 330 °C and 700 °C, respectively.

The samples used for Supplementary Fig. 9–11, Supplementary Figs. 13–14 and Supplementary Fig. 21 were prepared in the second batch, similar to the second batch preparation of $Fe_{1+\alpha}Te_2$. However, $NiCl_2 \cdot 6H_2O$ powder (35 mg, 99.9%, Sigma-Aldrich) and Se powder (3 g, 99.99%, Aladdin) were used as metal and chalcogen element precursors. The tube chamber was first purged by pure argon, and then the gas flow was maintained at 400 sccm Ar and 2 sccm $H_2$. The furnace zone was ramped to 200 °C over 30 min, then to 650 °C within 35 min, and then held for 10 min. The Se precursor was placed upstream at different distances from the edge of the furnace chamber, where the temperature at each position was recorded. After that, the furnace naturally cooled to room temperature.

## Growth of $V_{1+\varepsilon}Se_2$

As the melting point of $VCl_3$ is very low, metal precursor $VCl_3$ $_{1+\varepsilon2}$ (12 mg, 97%, Aladdin) in a quartz boat and Se powder (300 mg, 99.99%, Aladdin) in a quartz boat were both placed in the upstream temperature zone. Three pieces of mica substrates were set in the front of the downstream temperature zone in sequence (following the upstream zone). The temperature of the upstream zone and the downstream zone was ramped to 350 °C and 600 °C in 20 min and then retained for 10 min. Finally, the tube furnace cooled down to room temperature.

## Growth of $Cu_{1+\zeta}S_2$

A quartz tile with anhydrous $CuCl_2$ powder was put into the upstream temperature zone. S powder (150 mg, 99.99%, Macklin reagent) in a quartz boat was heated by a heating belt outside the furnace zone. Freshly exfoliated mica substrates were obliquely placed on quartz tile above the $CuCl_2$ powder. The temperature of the upstream area and heating belt were increased to 450 °C and 120 °C in 30 min, respectively. The growth period was about 10 min. Finally, the tube furnace was cooled down to room temperature.

## Growth of $Cr_{1+\sigma}Se_2$

The growth of $Cr_{1+\sigma}Se_2$ was the same as the growth of $Fe_{1+\alpha}Te_2$. Anhydrous $CrCl_3$ powder (5 mg, Sigma-Aldrich) and Se powder (300 mg, 99.99%, Aladdin) were employed as metal and chalcogen precursors, respectively. The temperature of the metal precursor and substrate zone was ramped to 750 °C in 35 min and then held for 10 min to synthesize materials. The temperature of the Se zone was set as 400 °C, 360 °C, 330 °C or 300 °C, depending on the requirements. The ramping and holding periods were 33 min and 12 min, respectively. Unless specified, the $Cr_{1+\sigma}Se_2$ samples used in the manuscript were prepared using a Se temperature of 330 °C.

## Sample transfer

The nonstoichiometric 2D nanoflakes on mica substrate can be transferred to other substrates or Cu grids with the assistance of polymethylmethacrylate (PMMA) film. In brief, nonstoichiometric 2D nanoflakes on mica were first spin-coated by a PMMA solution (4 wt.% in anisole) at a speed of 4000 rpm for 1 min. Then, the substrates were dipped into boiling DI water for 1 min after annealing at 80 °C for 5 min. PMMA film was then manually peeled off from the mica substrate. The floating PMMA film was fished by another substrate or Cu grids followed by 80 °C baking for 5 min. Finally, the PMMA was removed by hot acetone (50 °C), and then naturally dried in air at room temperature.

## Characterization

AFM, CFM and PFM characterizations were conducted on Asylum Research MFP-3D and Cypher S instruments at room temperature under ambient air. Silicon tips with a force constant of 2.8 N/m and Pt/Ir conductive coating were utilized for CFM and PFM characterizations. The drive voltage used for piezoelectric response measurement and ferroelectric domain imaging is AC voltage, which is switched between positive and negative during PFM scanning to detect the piezoelectric vibration and phase change of the sample, while the drive voltage in the hysteresis loop or for the lithography is DC voltage to reverse polarization direction. For lithography, ferroelectric hysteresis loop and electrical curve measurements, nonstoichiometric 2D nanoflakes were transferred on Cr/Au (5/50 nm) coated Si substrate and then grounded by connecting a conductive cable via the adhesion of silver paste. Each I-V curve in CFM is generated by averaging five-cycle I-V curves to exclude potential local instability or disturbance. XPS spectra were collected on a Thermo Fisher Scientific Nexsa G2 X-ray photoelectron spectrometer (XPS) equipment. TEM images, EDS spectra and SAED patterns were gathered on a thermal field emission transmission electron microscope instrument under an operation voltage of 200 kV

(JEOL JEM-2100F). STEM-HAADF and iDPC imaging were implemented on an aberration-corrected STEM microscope (Spectra 300, Thermofisher equipped with a field emission gun). The convergence half-angle of 29.9 mrad was used, and the collection half-angle of the HAADF detector ranged from 57 to 200 mrad for atomic-resolution observations. For a clear view, the image in (**h**) was filtered by an average background subtraction filter (ABSF) on Digital Micrograph software (v 3.53, Gatan Inc., USA). Magnetism hysteresis loops were collected on a physical property measurement system (Quantum Design) equipped with a vibrating sample magnetometer (VSM). Raman spectra were measured using a confocal Raman spectrometer using a laser source of 532 nm and 633 nm (Witec alpha300 R). SHG mapping was excited by 900 nm laser pulses on a Leica TCS SP8 MP Multiphoton/Confocal Microscope equipped with a sapphire compact laser system (Mai Tai HP Ti).

## Device fabrication and measurements
A physical masking method fabricated the two-terminal devices of nonstoichiometric 2D nanoflakes. The channel was firstly sheltered by 10 μm tungsten wires and fixed. Subsequently, Cr/Au (5/50 nm) electrodes were deposited using a Denton E-beam deposition system. To enable bending of the nonstoichiometric 2D $Cr_{1+o}Se_2$ nanoflakes transferred on polyimide (PI), the electrodes were connected out to 0.1 mm Cu wires by high-conductive silver glue (H20E, Epoxy Technology). The electrical measurements were done on a probe station system with a Keithley 4200a-SCS parameter analyzer at room temperature and atmospheric pressure.

## Reporting summary
Further information on research design is available in the Nature Portfolio Reporting Summary linked to this article.

# Data availability
The authors declare that all data supporting the findings of this study are available within the paper and Supplementary Information files.

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

## Acknowledgements
S.P.L. acknowledges support by grants from the Research Grants Council of the Hong Kong Special Administrative Region, China (grant nos. 15306321 and AoE/P-701/20) and PolyU (grant no. 1-YY5U). B.K.T. acknowledges support from the Ministry of Education, Singapore, under grant AcRF TIER 2- MOE-T2EP50121.

## Author contributions
S.P.L. and Y.H. conceived this work. Y.H. prepared materials. Y.H. and L.R. performed PFM characterizations and analysis. Y.H., L.Z. and F.S. performed XPS and Raman characterizations and analysis. Y.H., W.W., H.D. and S.C. performed EDS, TEM and STEM characterizations and analysis. Y.H. and S.P.L. wrote the manuscript with input from S.C. and B.K.T. All authors discussed the results and commented on the manu-script. S.C., B.K.T. and S.P.L. supervised the project.

## Competing interests
The authors declare no competing interests.
