## [Peer Review File · Nature Communications]

Extendable Piezo/ferroelectricity in Nonstoichiometric 2D Transition Metal DichalcogenidesREVIEWER COMMENTS

Reviewer #1 (Remarks to the Author):

Hu et al. reported a study on nonstoichiometric engineering of centrosymmetric 2D transition metal dichalcogenides with the aim to prove a new approach to explore piezoelectricity and ferroelectricity in van der Waals materials. To prove the validation of their strategy, the authors carried out abundant experimental measurements by using piezo force microscopy (PFM), second harmonic generation (SHG) technique, high-resolution transmission electron microscopy (HRTEM), and transport study in electronic device. The results and data are rich. However, to support their conclusion, several inconsistent issues need to be carefully clarified.

1. The main finding in this work is the emergence of piezoelectricity in intrinsically centrosymmetric TMDs via nonstoichiometric engineering. To determine the piezoelectric properties of the as-prepared sample, the authors performed PFM measurements. However, in the PFM amplitude images as shown in Fig. 2-3, the piezo force responses from these samples are quite scattered, which hints the intercalation of metal atoms between layers might be localized. Meanwhile, these PFM results cannot exclude the possible origin of these polar states from intralayer defects. Additionally, the piezoelectric coefficient is measured via multiple different driving voltages. To make the d_{33} more credible, the authors should consider the spatial variation of the PFM amplitudes under different driving voltages. Error bars should be added in Fig. 3b-d.

2. It is quite puzzled that the authors can still measure the piezo force response on a 20 nm Au coated $\text{Cr}_{1+\sigma}\text{Te}_2$ sample (see supplementary Fig. 4). When the PFM functions, the stimulating AC voltage needs to be applied between the tip and samples. The tip itself serves as one of the electrodes and the sample or the substrate should be grounded. If the sample is coated by metals, the stimulating AC voltage might not be applied onto the target samples, resulting in invalid measurements. I would like to suggest the authors to elaborate on this point.

3. For identifying the ferroelectricity in $\text{Fe}_{1+\sigma}\text{Te}_2$, a concern is that whether the observation of PFM loop and the artificial domain imaging as shown in Fig. 3e-g really establish that nonstoichiometric $\text{Fe}_{1+\sigma}\text{Te}_2$ thin films are ferroelectric, given caveats expressed, for example, in a recent review paper (Nature Communications 10:1661 (2019)). I would like to suggest the authors to provide more solid evidences, such as the observation of clear domain walls in the PFM amplitude images.

4. In the two-terminal devices of $\text{Fe}_{1+\sigma}\text{Te}_2$, the authors showed in-plane current hysteresis loop in Fig. 5f and attributed such behavior to the existence of in-plane ferroelectricity. Such understandings seem to be strongly contradictory to the results from PFM and HRTEM measurements as presented in Fig. 3 and Fig. 4. For PFM study, if robust, the demonstrated ferroelectricity should be out-of-plane. From the atomic structure revealed by HRTEM in Fig. 4, the authors found sliding of van der Waals layers and also the displacement of interlayer metal atoms, which also suggest the out-of-plane electric polarizations. Therefore, I would like to remind the authors on this point. It would be better to clarify this statement or understanding logically.

Reviewer #2 (Remarks to the Author):

Recommendation: Publish after major revisions noted.

The authors constructs nonstoichiometric 2D TMDs by introducing interlayer metals or intralayer vacancies in layered materials, which demonstrate ubiquitous piezo/ferroelectricity. The results are interesting and appealing for research fellows working on ferroelectric materials. However, there exist some issues that need to be addressed before its acceptance:

- 1、 The author mentions that $\text{Fe}_{1+\alpha}\text{Te}_2$ and $\text{Cu}_{1+\zeta}\text{S}_2$ demonstrated switchable spontaneous polarization. Then, what is the Curie temperature of the ferroelectric phase transition? It is suggested to conduct a temperature test for SHG to determine the specific temperature at which the phase transition occurs. Is ferroelectricity stable at room temperature?
- 2、 Figure 3a displays images of nonstoichiometric $\text{Cr}_{1+\sigma}\text{Se}_2$ nanoflake under positive drive voltages. It would be beneficial to also showcase the results under negative voltage. Additionally, in Figures 3f and 3g, it is necessary to exclude the influence of morphology on the intrinsic ferroelectric domain distribution.
- 3、 The ferroelectric reversal observed in Figure 3e does not appear to be a complete 180° change. Additionally, it would be beneficial to provide an atomic-level explanation for the intrinsic reasons behind ferroelectric reversal.
- 4、 Do other materials exhibit a similar trend in piezoelectric coefficients with changes in chemical composition as observed in $\text{Cr}_{1+\sigma}\text{Se}_2$? It would be advantageous to provide a more detailed and physics-based explanation of how variations in composition impact the structure and piezoelectric coefficients.
- 5、 The authors are suggested to provide a comparison between the in-plane and out-of-plane results for the magnetic hysteresis loops shown in Figure 5e.

Reviewer #3 (Remarks to the Author):

In this work, Shu Ping Lau. et al proposed 2D novel metal intercalated $\text{Fe}_{1+\alpha}\text{Te}_2$, $\text{Co}_{1+\beta}\text{S}_2$, $\text{Mn}_{1+\gamma}\text{Se}_2$, $\text{Ni}_{1+\delta}\text{Se}_2$, $\text{V}_{1+\epsilon}\text{Se}_2$, $\text{Cu}_{1+\zeta}\text{S}_2$, and $\text{Cr}_{1+\sigma}\text{Se}_2$, in which the atomic-scale microstructure of interlayer metal atoms was revealed using STEM-iDPC. The piezoelectric performance of representative 2D TMDs was confirmed through PFM, while the dielectric or electrostatic effects or flexoelectricity were excluded. Their piezoelectric coefficients were found to be correlated with the thickness and chemical composition. The paper is well organized and the idea is novel. I recommend this high-quality work be published in Nature Communications after addressing the following two comments.

1. The out-of-plane (OP) piezoelectricity of nonstoichiometric 2D TMDs was strong, while the in-plane (IP) PFM mapping images showed weak and unstable piezoelectric responses (Supplementary Figure 5). In Figure 5, the authors demonstrated a familiar two-terminal flexible device to manifest its applications in nanogenerators, memristors, and synapses. The release and stretch processed both contain vertical and lateral deformation in a micro lateral size crystal, bringing vertical and lateral stress on nonstoichiometric 2D TMDs. According to the above PFM results, vertical direction stress (out-of-plane, d_{33}) contributed a large quantity of piezoelectricity-generated electrons. However, the electrodes of the two terminal devices were lateral direction (IP, $d_{11}/d_{22}/d_{21}$), indicating that the electrons mainly came from IP piezoelectricity (Figure 5a and 5f). Therefore, I suggest the authors fabricate an up-down two-terminal electrode device to be consistent with its OP piezoelectricity.

Otherwise, the d31 direction piezoelectricity should be carried out and studied systematically.

2. In Supplementary Table 2, the d33 of nonstoichiometric 2D Cr_{1+σ}Se₂ is 0.71 ± 0.02 pm/V. Compared with other reported materials, the performance is not outstanding. Is there room for improvement?

We are pleased to answer the Reviewer's questions point-by-point as follows:

Reviewer #1 (Remarks to the Author):

Hu et al. reported a study on nonstoichiometric engineering of centrosymmetric 2D transition metal dichalcogenides with the aim to prove a new approach to explore piezoelectricity and ferroelectricity in van der Waals materials. To prove the validation of their strategy, the authors carried out abundant experimental measurements by using piezo force microscopy (PFM), second harmonic generation (SHG) technique, high-resolution transmission electron microscopy (HRTEM), and transport study in electronic device. The results and data are rich. However, to support their conclusion, several inconsistent issues need to be carefully clarified.

1. The main finding in this work is the emergence of piezoelectricity in intrinsically centrosymmetric TMDs via nonstoichiometric engineering. To determine the piezoelectric properties of the as-prepared sample, the authors performed PFM measurements. However, in the PFM amplitude images as shown in Fig. 2-3, the piezo force responses from these samples are quite scattered, which hints the intercalation of metal atoms between layers might be localized. Meanwhile, these PFM results cannot exclude the possible origin of these polar states from intralayer defects. Additionally, the piezoelectric coefficient is measured via multiple different driving voltages. To make the d_{33} more credible, the authors should consider the spatial variation of the PFM amplitudes under different driving voltages. Error bars should be added in Fig. 3b-d.

Response: We appreciate your meticulous review and valuable suggestions. Some of the PFM amplitude images indeed show a non-uniform distribution of signals. As per your suggestion, the non-uniform signals may originate from the localized metal atoms intercalation and non-uniform intralayer defects. Besides, the ferroelectric domain switching and rough surface topography may also contribute to the inhomogeneous amplitude signals. Moreover, the polar states represented by PFM results may also originate from intralayer defects. Thus, to clarify the description, we added some related discussions to the manuscript.

In the manuscript, the intrinsic amplitude value used to calculate the piezoelectric coefficient was statistically extracted from the intrinsic amplitude image, as marked as the blue area for sample amplitude and the red area for substrate amplitude in **Supplementary Figure 5a**. The effective piezoelectric coefficient d_{33} was also recalculated after including error bars. The effective d_{33} changes from 0.71 ± 0.02 pm/V to 0.65 ± 0.12 pm/V after accounting for the standard deviation (**Figure 3b**), giving a more scientific expression of the effective d_{33} .

Then, a more detailed region cutting is also proposed to consider the spatial variation of the PFM amplitude and evaluate the spatial deviation of the piezoelectric coefficient. As illustrated in **Supplementary Figure 5b-c**, three regions are divided and the corresponding effective d_{33} was separately calculated. The differences in the resulted mean effective d_{33} value for each region (0.61 ± 0.12 pm/V, 0.67 ± 0.14 pm/V and 0.63

± 0.10 pm/V) are in the second significant digit, and within the margin of standard error. Moreover, the average value of the effective d_{33} of three different regions is about 0.64 pm/V, which is very close to effective d_{33} (0.65 ± 0.12 pm/V) extracted from the total blue region (Supplementary Figure 5a and Figure 3b). Especially, the differences are acceptable after adding the standard deviation error. Thus, the effective piezoelectric in the paper will be calculated based on the amplitude of the total area of the nanoflake. Besides, significant standard deviations will be added to guarantee a more scientific and accurate expression.

Error bars for all plot graphs related to the amplitudes and piezoelectric coefficients are also added to exhibit their deviations.

Related discussions and modifications were made in the **Manuscript** and **Supplementary information** as follows:

“However, some of those amplitude images exhibit a non-uniform distribution of signals, which may originate from the localized nonstoichiometricity of metal atoms intercalation and uneven intralayer defects. Besides, the ferroelectric domain switching and rough surface topography may also contribute to the inhomogeneous amplitude signals.”

“Those detailed electron microscopic analyses distinctly uncover that the formation mechanism of nonstoichiometric 2D compounds is mostly attributed to atomic-level interlayer metal intercalations between sandwich MX₂ layers and potential slight intralayer defects.”

“In addition to interlayer sliding, intrinsic structural distortion induced by asymmetrical metal coordination and potential intralayer defects also accelerate the occurrence of lateral electric dipoles and yield stable IP polarization^{63–65}. Switching of the external electrical field could control the lateral displacement of the intercalated metal ions and intralayer defects, thereby switching the direction of IP spontaneous polarization in nonstoichiometric 2D TMDs.”

“2.1 Calculation of effective d_{33} and $d_{31/32}$ ”

A single frequency PFM mode was used to reveal in-plane piezoelectric response (Supplementary Figure 12), while dual AC resonance tracking (DART) PFM with lateral and vertical modes were used to measure IP and OOP ferroelectric domains, piezoelectric response and corresponding piezoelectric coefficient. In the DART OOP and IP modes, two frequencies near the center of the intrinsic resonance peak were chosen to collect two sets of amplitude and phase images (Supplementary Figure 3). However, those amplitudes are amplified by resonance rather than real/intrinsic amplitudes. The real/intrinsic amplitudes are then calculated by a simple harmonic oscillator model (SHO) (such as the bottom planes in Figure 3a and 3e). The grey spots indicate missing data which is unsuccessfully calculated during SHO calculation. The mean amplitude value and corresponding standard deviation for the interested region in the sample or substrate are exported by performing a Gaussian fit on the statistical diagram of all site values of the amplitude image. By linear fitting intrinsic amplitude to the drive AC voltage under consideration of the error bars, the mean value of effective

piezoelectric coefficient (slope) and standard deviation are recorded (Figure 3b,f and Supplementary Figure 5,10,13). The value of the effective IP piezoelectric coefficient ($d_{31/32}$) is finally calibrated by a standard z-cut LN crystal.

A detailed region-cutting method was conducted to take account of the spatial variation of PFM amplitudes and evaluate the deviation of the piezoelectric coefficient in space. As illustrated in Supplementary Figure 5b,c, the intrinsic amplitude image of nanoflake in Supplementary Figure 5a acquired at a drive AC voltage of 4 V was divided into three regions, and the corresponding effective d_{33} was separately calculated through the method mentioned above. The calculated effective d_{33} values of area I, II and III were 0.61 ± 0.12 pm/V, 0.67 ± 0.14 pm/V and 0.63 ± 0.10 pm/V, respectively. The value differences between the three effective d_{33} are only in the second significant digit and are within the margin of standard error. Moreover, the average value of the effective d_{33} of three different regions is about 0.64 pm/V, which is very close to the effective d_{33} (0.65 ± 0.12 pm/V) counted from the total blue region (Supplementary Figure 5a and Figure 3b). Although the local non-uniform distribution of amplitude signals will produce different effective d_{33} , the effective d_{33} calculated from the signal in the entire nanoflake region can be used to represent the piezoelectric coefficient of the nanoflakes.”

“Thus, by fitting multiple points of intrinsic piezoelectric amplitude of the sample to the drive AC voltages, the effective d_{33} of a 5.6 nm thick nonstoichiometric 2D $\text{Cr}_{1+\sigma}\text{Se}_2$ nanoflake and a 47.8 nm thick nonstoichiometric 2D $\text{Ni}_{1+\delta}\text{Se}_2$ were counted to be 0.65 ± 0.12 pm/V and 6.78 ± 0.6 pm/V, respectively (Figure 3b, more details are elaborated in Supplementary Note 2 and Supplementary Figure 5–11).”

Supplementary Figure 5. (a) AFM height image and corresponding profile curve of a nonstoichiometric 2D $\text{Cr}_{1+\sigma}\text{Se}_2$ triangular nanoflake. (b) Corresponding amplitude image of nanoflakes in (a) driven by AC voltage of 4 V. Red and blue (purple) shadow masks in (a) and (b) represent regions that are used to extract substrate and sample amplitude statistic diagram for the calculation of effective d_{33} calculation. (c) The intrinsic amplitudes collected from valid areas of I, II and III as a function of the drive AC voltage curve.

Fig. 3 | Tuning piezoelectricity of nonstoichiometric $\text{Cr}_{1+\sigma}\text{Se}_2$ nanoflakes and emerging ferroelectricity in nonstoichiometric $\text{Fe}_{1+\alpha}\text{Te}_2$ nanoflakes. **a** OOP amplitude images (up), phase images (middle) and intrinsic amplitude images (bottom) of a nonstoichiometric $\text{Cr}_{1+\sigma}\text{Se}_2$ nanoflake under different drive voltages. **b** Corresponding OOP amplitude evolution curve with drive AC voltage. Orange, pink, blue and prasinous lines are assigned to resonance-amplified OOP amplitude of the sample, resonance-amplified OOP amplitude of the substrate, original intrinsic OOP amplitude of the sample and linearly fitted intrinsic OOP amplitude of the sample as a function of drive AC voltage, respectively. **c** Evolution of resonance-amplified OOP amplitudes of nonstoichiometric $\text{Cr}_{1+\sigma}\text{Se}_2$ nanoflakes with different thicknesses. **d** Effective d_{33} of nonstoichiometric $\text{Cr}_{1+\sigma}\text{Se}_2$ nanoflakes with different values of σ . **e** Height image and different AC voltage-driven intrinsic amplitude images of a nonstoichiometric $\text{Cr}_{1+\sigma}\text{Se}_2$ nanoflake. **f** Corresponding intrinsic IP amplitude in **e** as a function of the drive AC voltage of substrate and nanoflake sample. **g** Local OOP ferroelectric switching spectra under DC off state of a nonstoichiometric $\text{Fe}_{1+\alpha}\text{Te}_2$ nanoflake. **h** Phase images of a nonstoichiometric $\text{Fe}_{1+\alpha}\text{Te}_2$ nanoflake after sequential lithography by using different patterns. **i** Local IP ferroelectric hysteresis loops under DC off state of a nonstoichiometric $\text{Fe}_{1+\alpha}\text{Te}_2$ nanoflake. The black and white dotted line

frames indicate +6 V and –6 V lithography area, respectively. **j-k** IP amplitude (**j**) and IP phase (**k**) images of a nonstoichiometric $\text{Fe}_{1+\alpha}\text{Te}_2$ nanoflake after electrical field writing along two perpendicular lines. The black line and white line indicate first +6 V and then –6 V lithography.

2. It is quite puzzled that the authors can still measure the piezo force response on a 20 nm Au coated $\text{Cr}_{1+\sigma}\text{Se}_2$ sample (see supplementary Fig. 4). When the PFM functions, the stimulating AC voltage needs to be applied between the tip and samples. The tip itself serves as one of the electrodes and the sample or the substrate should be grounded. If the sample is coated by metals, the stimulating AC voltage might not be applied onto the target samples, resulting invalid measurements. I would like to suggest the authors to elaborate on this point.

Response: Thank you for the very professional questions. As you stated intrinsic mechanism of PFM measurement, the PFM should avoid short circuits between tip and substrate, and we apologize for the ambiguity we have caused. The substrate used for PFM measurement of Au coated $\text{Cr}_{1+\sigma}\text{Te}_2$ nanoflakes is insulator mica. Due to the sharp conductive tip, the small contact area between the tip and sample may lead to charge accumulation and local sample rupture. Besides, flexoelectricity and electrostatic effect triggered by the uneven distribution of drive AC voltage may also induce pseudo-piezoresponse. Moreover, the electrochemical reaction facilitated by the water meniscus at the tip-sample junction may also shelter the real piezoelectric response. Thus, the top Au layer can act as a top electrode to apply AC voltage on the sample and simultaneously isolate the sample surface from surrounding ambient air and water. As a result, no noticeable differences were observed before and after Au deposition, indicating real piezoelectricity in nonstoichiometric 2D TMDs.

To make the statement clearer, we added some text to the relevant sections.

“However, confirmation of the detected signal due to intrinsic piezo/ferroelectricity is decisive to rule out spurious piezo/ferroelectric response arising from carrier injection, flexoelectricity, morphology-induced friction and electrochemical reactions promoted by the water meniscus effect⁵⁵.”

“Beyond, a 20 nm thick Au film was deposited on a 3.6 nm nonstoichiometric 2D $\text{Cr}_{1+\sigma}\text{Se}_2$ nanoflake on mica substrate to exclude potential artifacts triggered by surface charging, flexoelectricity or air-environmental electrochemical reaction (Supplementary Figure 4).”

Supplementary Figure 4. (a) AFM image, (b) height profile, and (c) OOP PFM mapping images of a nonstoichiometric $\text{Cr}_{1+\sigma}\text{Se}_2$ triangular nanoflake on the mica substrate after the deposition of 20 nm thick Au film. Au deposition can weaken the charging and electrostatic effects, increase the uniformity of the applied electric field and avoid the flexoelectric effect. It is indicated that the surface morphology of the sample becomes rougher after Au deposition, while the piezoelectric effect is still apparent, confirming the real piezoelectricity of the nanoflakes.

3. For identifying the ferroelectricity in $\text{Fe}_{1+\sigma}\text{Te}_2$, a concern is that whether the observation of PFM loop and the artificial domain imaging as shown in Fig. 3e-g really establish that nonstoichiometric $\text{Fe}_{1+\sigma}\text{Te}_2$ thin films are ferroelectric, given caveats expressed, for example, in a recent review paper (Nature Communications 10:1661 (2019)). I would like to suggest the authors to provide more solid evidences, such as the observation of clear domain walls in the PFM amplitude images.

Response: Thanks for your professional and particular suggestion. The phase and amplitude images of $\text{Fe}_{1+\alpha}\text{Te}_2$ nanoflakes in Figure 2b demonstrate contrasting signals, which may indicate ferroelectric domains. However, the domain is ambiguous as the low resolution. Thus, we investigated the magnified region of $\text{Fe}_{1+\alpha}\text{Te}_2$ nanoflakes to obtain high-resolution OOP and IP amplitude and phase images. As illustrated in Supplementary Figure S15, both OOP and IP phase images show strong contrasting signals with sharp boundaries, which can be attributed to ferroelectric domains with opposite polarization directions. Despite the rough morphology of the nanoflakes, both amplitude and phase images are significantly independent of the height image. The ferroelectric domains of $\text{Fe}_{1+\alpha}\text{Te}_2$ nanoflakes are not in regular stripes like those of bulk

ferroelectric crystals, but are similar to other mechanically exfoliated 2D materials, such as CuInP_2S_6 and In_2Se_3 (*Nat. Commun.* 7(1), 1–6. *Adv. Funct. Mater.* 2020, 30, 2004609.). The OOP and IP PFM amplitude and phase of nonstoichiometric 2D $\text{Fe}_{1+\alpha}\text{Te}_2$ as a function of the DC voltage also exhibit typical butterfly shapes and hysteresis loops (Figure 3g,i), confirming its ferroelectric features.

Furthermore, to further confirm the exotic ferroelectric behavior, the OOP polarization lithography was undertaken by applying ± 6 V DC voltage on $\text{Fe}_{1+\alpha}\text{Te}_2$ (white and black dotted lines in Figure 3h). Subsequent PFM scanning presents two contrast areas in the phase image, suggesting the large-area switchable spontaneous polarization of nonstoichiometric 2D $\text{Fe}_{1+\alpha}\text{Te}_2$ (up left in Figure 3h). After sequential electrical field writing of different patterns and then conducting PFM imaging, the ferroelectric domains of $\text{Fe}_{1+\alpha}\text{Te}_2$ nanoflakes correspondingly switch, showing distinct signal differences even after 4 times of lithography (Figure 3h). Besides, the switching of IP spontaneous polaritons of nonstoichiometric 2D $\text{Fe}_{1+\alpha}\text{Te}_2$ nanoflakes was also verified by sequentially scanning a black line with 6 V and a white line with -6 V voltage along the direction of the arrow in Figure 3j,k. The first lithography with a positive voltage (black line) shows obvious manipulation of the polarization direction (Supplementary Note 4, Figure 3k), and the subsequent negative voltage line, in turn, changes the direction of ferroelectric polarization (as displayed in the intersection in Figure 3k). The multiple lithography of ferroelectric domains suggests rewritable and robust polarization of nonstoichiometric 2D $\text{Fe}_{1+\alpha}\text{Te}_2$ nanoflakes.

By combining the characterizations of high-resolution ferroelectric domains, PFM hysteresis loops and multiple artificial lithography, it can be determined that the spontaneous polarization mostly originated from the intrinsic ferroelectricity of the nonstoichiometric nanoflakes rather than external spurious response (*Nat. Commun.* 10, 1661 (2019).). The spontaneous polarization of nonstoichiometric 2D nanoflakes reveals that nonstoichiometric ratio engineering is an effective approach for accessing the ferroelectricity of 2D materials.

Relevant PFM characterizations and discussions were added as follows:

“The magnified amplitude and phase images collected in IP and OOP modes demonstrate very distinct and sharp domains, suggesting the potential ferroelectricity of $\text{Fe}_{1+\alpha}\text{Te}_2$ (Supplementary Figure 15, Supplementary Note 4).”

“To further confirm the exotic ferroelectric behavior, the OOP polarization lithography was undertaken by applying ± 6 V DC voltage on $\text{Fe}_{1+\alpha}\text{Te}_2$ (white and black dotted lines in Figure 3h). Subsequent PFM scanning presents two contrast areas in the phase image, suggesting the large-area switchable spontaneous polarization of nonstoichiometric 2D $\text{Fe}_{1+\alpha}\text{Te}_2$ (up left in Figure 3h). After sequential electrical field writing of different patterns and then conducting PFM imaging, the ferroelectric domains of $\text{Fe}_{1+\alpha}\text{Te}_2$ nanoflakes correspondingly switch, showing distinct signal differences even after 4 times of lithography (Figure 3h). Besides, the switching of IP spontaneous polaritons of nonstoichiometric 2D $\text{Fe}_{1+\alpha}\text{Te}_2$ nanoflakes was also verified by sequentially scanning a black line with 6 V and a white line with -6 V voltage along the direction of the arrow in Figure 3j,k. The first lithography with a positive voltage (black line) shows obvious

manipulation of the polarization direction (Supplementary Note 4, Figure 3k), and the subsequent negative voltage line, in turn, changes the direction of ferroelectric polarization (as displayed in the intersection in Figure 3k). The multiple lithography of ferroelectric domains suggests rewritable and robust polarization of nonstoichiometric 2D $\text{Fe}_{1+\alpha}\text{Te}_2$ nanoflakes.”

“Especially, morphology-independent phase and amplitude images of nonstoichiometric nanoflakes exclude the pseudo-ferroelectricity originating from surface topography or oxidation/damage (Supplementary Figure 15, Supplementary Figure 16c, Supplementary Figure 17 and Supplementary Figure 18b).”

“By combining the characterizations of high-resolution ferroelectric domains, PFM hysteresis loops and multiple artificial lithography, it can be determined that the spontaneous polarization mostly originated from the intrinsic ferroelectricity of the nonstoichiometric nanoflakes rather than external spurious response⁵⁵.”

“In the case of samples prepared in the second batch, the experimental parameters and instruments are detailed accordingly. All of the samples prepared in the second batch will be noted.”

“The samples used in Supplementary Figure 15d-f were prepared in the second batch. The experiments were prepared in a 2-inch tube furnace with a single heating zone. Te particles (6 g, 99.99%, Sigma-Aldrich) were placed in a quartz boat upstream and anhydrous FeCl_2 powders (4.5 mg, 98%, Sigma-Aldrich) were set in a quartz tile in the center of the tube furnace zone, respectively. Freshly exfoliated mica was used as substrate which was faced down the FeCl_2 powder. The tube chamber was firstly purged by 100 sccm Ar and 90 sccm H_2 gases for 30 min and then the flow was maintained. Subsequently, the furnace zone was heated to 530 °C within 30 min and then maintained for 10 min. The temperature of the Te precursor was estimated to be ~475 °C during the material growth stage. After that, the furnace naturally cooled to room temperature.”

“AFM, CFM and PFM characterizations were conducted on Asylum Research MFP-3D and Cypher S instruments at room temperature under ambient air. Silicon tips with a force constant of 2.8 N/m and Pt/Ir conductive coating were utilized for CFM and PFM characterizations. For lithography, ferroelectric hysteresis loop and electrical curve measurements, nonstoichiometric 2D nanoflakes were transferred on Cr/Au (5/50 nm) coated Si substrate and then grounded by connecting a conductive cable via the adhesion of silver paste.”

Supplementary Figure 15. (a-c) Magnified height image, amplitude image and phase image of a $\text{Fe}_{1+a}\text{Te}_2$ nanoflake obtained by OOP PFM mode. (d-f) Height image, amplitude image and phase image of the corner of a $\text{Fe}_{1+a}\text{Te}_2$ nanoflake collected by IP PFM mode. Despite the rough topography of the nanoflakes, both amplitude and phase images are significantly independent of the height images.

Fig. 3 | Tuning piezoelectricity of nonstoichiometric $\text{Cr}_{1+\sigma}\text{Se}_2$ nanoflakes and emerging ferroelectricity in nonstoichiometric $\text{Fe}_{1+\alpha}\text{Te}_2$ nanoflakes. **a** OOP amplitude images (up), phase images (middle) and intrinsic amplitude images (bottom) of a nonstoichiometric $\text{Cr}_{1+\sigma}\text{Se}_2$ nanoflake under different drive voltages. **b** Corresponding OOP amplitude evolution curve with drive AC voltage. Orange, pink, blue and prasinous lines are assigned to the resonance-amplified OOP amplitude of the sample, resonance-amplified OOP amplitude of the substrate, original intrinsic OOP amplitude of the sample and linearly fitted intrinsic OOP amplitude of the sample as a function of drive AC voltage, respectively. **c** Evolution of resonance-amplified OOP amplitudes of nonstoichiometric $\text{Cr}_{1+\sigma}\text{Se}_2$ nanoflakes with different thicknesses. **d** Effective d_{33} of nonstoichiometric $\text{Cr}_{1+\sigma}\text{Se}_2$ nanoflakes with different values of σ . **e** Height image and different AC voltage-driven intrinsic amplitude images of a nonstoichiometric $\text{Cr}_{1+\sigma}\text{Se}_2$ nanoflake. **f** Corresponding intrinsic IP amplitude in **e** as a function of the drive AC voltage of substrate and nanoflake sample. **g** Local OOP ferroelectric switching spectra under DC off state of a nonstoichiometric $\text{Fe}_{1+\alpha}\text{Te}_2$ nanoflake. **h** Phase images of a nonstoichiometric $\text{Fe}_{1+\alpha}\text{Te}_2$ nanoflake after sequential lithography by using different patterns. **i** Local IP ferroelectric hysteresis loops under DC off state of a nonstoichiometric $\text{Fe}_{1+\alpha}\text{Te}_2$ nanoflake. The black and white dotted line

frames indicate +6 V and –6 V lithography area, respectively. **j-k** IP amplitude (**j**) and IP phase (**k**) images of a nonstoichiometric $\text{Fe}_{1+\alpha}\text{Te}_2$ nanoflake after electrical field writing along two perpendicular lines. The black line and white line indicate first +6 V and then –6 V lithography.

Supplementary Figure 17. (a-d) Hight images of a nonstoichiometric $\text{Fe}_{1+\alpha}\text{Te}_2$ nanoflake after sequential lithography with different patterns. (e-h) Corresponding amplitude images of (a-d). (i) AFM height image of a nonstoichiometric $\text{Fe}_{1+\alpha}\text{Te}_2$ nanoflake on which IP ferroelectric lithography is performed.

4. In the two-terminal devices of $\text{Fe}_{1+\sigma}\text{Te}_2$, the authors showed in-plane current hysteresis loop in Fig. 5f and attributed such behavior to the existence of in-plane ferroelectricity. Such understandings seem to be strongly contradictory to the results from PFM and HRTEM measurements as presented in Fig. 3 and Fig. 4. For PFM study, if robust, the demonstrated ferroelectricity should be out-of-plane. From the atomic structure revealed by HRTEM in Fig. 4, the authors found sliding of van der Waals layers and also the displacement of interlayer metal atoms, which also suggest the out-of-plane electric polarizations. Therefore, I would like to remind the authors on this point. It would be better to clarify this statement or understanding logically.

Response: Thank you for the kind reminder. We deeply understand the concern about inconsistency between the IP devices and OOP PFM characterizations. Thus, we have made a lot of extra efforts to comprehensively investigate and clarify the IP and OOP ferroelectricity of $\text{Fe}_{1+\sigma}\text{Te}_2$, including IP and OOP high-resolution domain imaging (Supplementary Figure S15), OOP sequential lithography (Figure 3h), IP PFM hysteresis loops (Figure 3i), IP domain lithography (Supplementary Note 4, Figure 3j-k) and I-V electrical curve along OOP direction (Figure 5e).

From the perspective of IP and OOP PFM ferroelectricity, $\text{Fe}_{1+\sigma}\text{Te}_2$ nanoflakes show both IP and OP ferroelectric domains and reversible polarization, as discussed in the **Question 3** section. The occurrence of lateral electric dipoles and yielding of stable IP polarization may be attributed to the intrinsic structural distortion induced by asymmetrical metal coordination and potential intralayer defects (*Nat. Rev. Mater.* **8**, 25-40 (2023)). Switching of the external electrical field could control the lateral

displacement of the intercalated metal ions and intralayer defects, thereby switching the direction of IP spontaneous polarization in nonstoichiometric 2D TMDs. The manifestation of IP ferroelectricity in non-stoichiometric $\text{Fe}_{1+\sigma}\text{Te}_2$ nanoflakes makes it a candidate material for in-plane ferroelectric memristor devices.

Besides, we collected the I-V curves along the vertical (out-of-plane) direction of the nanoflake to study electrical behavior corresponding to OOP ferroelectricity. The I-V curves of the vertical device show typical loop windows at different ranges of DC voltage, confirming the reversal of the polarization states at more negative or positive voltages and thus changing the resistance of the nanoflakes (Figure 5e). As the voltage range increases, the loop window accordingly expands. The I-V characteristics substantially verify the OOP ferroelectricity, signifying the FTJ application (Figure 5e). However, higher voltage also potentially enhances the tunneling current and causes the loop deviation (as indicated by the I-V curve with a sweeping range of -10 V to 10 V in Figure 5e).

IP electrical transport of lateral two-terminal devices based on a 2D $\text{Fe}_{1+\alpha}\text{Te}_2$ nanoflake was also measured in different sweep ranges of voltage. The I-V curves of a 2D $\text{Fe}_{1+\alpha}\text{Te}_2$ nanoflake demonstrate a current hysteresis loop with typical high and low resistance states, confirming its IP ferroelectricity (Figure 5f and Supplementary Figure 26a).

In this way, a one-to-one logical relationship was formed by individually performing PFM characterizations and electrical characteristic measurements of the 2D $\text{Fe}_{1+\alpha}\text{Te}_2$ nanoflakes in the IP and OOP directions.

Relevant revisions have been made as follows:

“In addition to interlayer sliding, intrinsic structural distortion induced by asymmetrical metal coordination and potential intralayer defects also accelerate the occurrence of lateral electric dipoles and yield stable IP polarization^{63–65}. Switching of the external electrical field could control the lateral displacement of the intercalated metal ions and intralayer defects, thereby switching the direction of IP spontaneous polarization in nonstoichiometric 2D TMDs.”

“The IP and OOP ferroelectricity of the 2D $\text{Fe}_{1+\alpha}\text{Te}_2$ nanoflakes make them candidates for the ferroelectric tunnel junction (FTJ) and ferroelectric memristor applications. Firstly, the vertical electrical transport behaviors were investigated to elaborate OOP ferroelectricity via conductive force microscopy (CFM) (Figure 5e). The high voltages (more negative or positive) reverse the polarization direction of the ferroelectric nanoflakes, leading to the two different resistances associated with different I-V curve slopes (Figure 5e). The enlarging loop windows with the increase of voltage range substantially verify the OOP ferroelectricity and signify the FTJ application (Figure 5e). However, higher voltage also potentially enhances the tunneling current and causes the loop deviation (as indicated by the I-V curve with a sweeping range of -10 V to 10 V in Figure 5e).”

Supplementary Note 4. Elaboration of IP PFM characterization and lithography.

Lateral PFM (IP mode) can detect an in-plane component of the dipoles as the lateral motion of the cantilever due to bias-induced surface shearing. Owing to the small

contact area of the conductive tip on which drive AC voltage is applied, the revealed IP domains may not be as robust as the OOP one and the calculated effective piezoelectric coefficient is related to d_{31}/d_{32} . However, it is completely sufficient to qualitatively reveal the IP ferroelectric domains and calculate piezoelectric coefficients with reference to standard *z*-cut lithium niobate (LN) wafer. Moreover, angle-dependent IP PFM were performed to precisely ascertain the d_{31} and d_{32} component of IP piezoelectric response (Supplementary Figure 14).

Unlike switching the ferroelectric domain through patterns in the OOP lithography, two perpendicular scan lines with opposite voltages are employed to switch IP ferroelectric domains to avoid interaction between each scan line in the pattern (Figure 3j,k). Since the applied DC voltage is along the vertical direction, the electric field is a sphere centered on the tip. Besides, the diameter of the tip is very small with tens of nanometers. Thus, the dwell time of each point in the line scan is prolonged compared with pattern lithography. After line scan lithography, the polarization directions on both sides of the scan line are theoretically opposite and perpendicular to the polarization direction on the scan line path. However, the IP PFM can only detect the polariton components perpendicular to the tip cantilever. Therefore, the phase patterning along the transverse line (black line) is obvious, while the phase along the longitudinal direction shows the same degree as the two side areas after lithography (white line) (Figure 3k). However, the horizontal polarization direction is obviously changed after longitudinal scanning (as shown by the point of intersection in Figure 3k), strongly proving the reverse of IP polarization of nonstoichiometric 2D TMDs. In addition, the phase and amplitude of the substrate are almost unchanged after line lithography, demonstrating the phase change originates from the reversion of the IP polarization rather than the charge accumulation effect.”

“Each I-V curve in CFM is generated by averaging five-cycle I-V curves to exclude potential local instability or disturbance.”

Fig. 5 | Electrical behavior of two-terminal devices based on nonstoichiometric 2D materials. **a** Digit photo images and an optical image of a two-terminal nanogenerator based on a nonstoichiometric 2D $\text{Cr}_{1+\sigma}\text{Se}_2$ nanoflake. **b** Output voltage and short circuit current of a 2D $\text{Cr}_{1+\sigma}\text{Se}_2$ nanoflake under one release-stretch-release cycle. **c-d** Output voltage and short circuit current as a function of time under several cycles of tensile strain. **e** Electrical transport behaviors of the nonstoichiometric 2D $\text{Fe}_{1+\alpha}\text{Te}_2$ nanoflakes along OOP direction. The inset illustrates the principal scheme of a CFM using a conductive tip and Au layer as top and bottom electrodes, respectively. **f** I-V curves of a two-terminal nonstoichiometric 2D $\text{Fe}_{1+\alpha}\text{Te}_2$ nanoflake device with different maximum sweeping voltages. **g** I-V curves of a two-terminal nonstoichiometric 2D $\text{Fe}_{1+\alpha}\text{Te}_2$ nanoflake device with 109 cycle sweeping. The sweeping voltage is set in a range of -4 to 4 V. **h** Drain currents retention curves read at drain voltages of 10 and 0.5 V (up) and read at 2 V after 10 V and -10 V bias poling (below). **i** I-V curves after 7 times accumulation. The inset shows the enlarged grey dotted frame of the curves in logarithmic coordinates. Drain voltage sweeps from 0 V to 10 V in each cycle. **j** Currents and open circuit voltages evolution in the 7 cycles sweeping.

Thanks again for your professional comments, which greatly help us improve the manuscript. Hopefully, we've addressed all of your concerns.

Reviewer #2 (Remarks to the Author):

Recommendation: Publish after major revisions noted.

The authors construct nonstoichiometric 2D TMDs by introducing interlayer metals or intralayer vacancies in layered materials, which demonstrate ubiquitous piezo/ferroelectricity. The results are interesting and appealing for research fellows working on ferroelectric materials. However, there exist some issues that need to be addressed before its acceptance:

1, The author mentions that $\text{Fe}_{1+\alpha}\text{Te}_2$ and $\text{Cu}_{1+\zeta}\text{S}_2$ demonstrated switchable spontaneous polarization. Then, what is the Curie temperature of the ferroelectric phase transition? It is suggested to conduct a temperature test for SHG to determine the specific temperature at which the phase transition occurs. Is ferroelectricity stable at room temperature?

Response: Thanks for your insightful suggestion and question. Due to the facility limitations, we ask for your understanding that we conducted temperature-dependent Raman spectroscopy instead of SHG. Temperature-dependent Raman spectroscopy is also a powerful tool in determining phase transition and Curie temperature (*Science* 2006, 313(5793), 1614–1616.; *J. Am. Ceram. Soc.* 1996, 79(10), 2666–2672.). As illustrated in **Supplementary Figure 19**, both Raman peak intensity of $\text{Fe}_{1+\alpha}\text{Te}_2$ and $\text{Cu}_{1+\zeta}\text{S}_2$ nanoflakes fluctuate in a small range during heating from room temperature to 523 K, indicating strong stability of ferroelectric phase without phase transition and giving Curie temperatures higher than 523 K.

The nonstoichiometric 2D $\text{Fe}_{1+\alpha}\text{Te}_2$ nanoflake demonstrates superior ferroelectric stability even after 6 months of preservation in ambient air (**Figure 3g-k**). Furthermore, the sequential OOP polarization lithography was undertaken by applying ± 6 V DC voltage on $\text{Fe}_{1+\alpha}\text{Te}_2$ (white and black dotted lines in **Figure 3h**). Subsequent PFM scanning presents two contrast areas in the phase image, suggesting the large-area switchable spontaneous polarization of nonstoichiometric 2D $\text{Fe}_{1+\alpha}\text{Te}_2$ (up left in **Figure 3h**). After sequential electrical field writing of different patterns and then conducting PFM imaging, the ferroelectric domains of $\text{Fe}_{1+\alpha}\text{Te}_2$ nanoflakes correspondingly switched and presented distinct signal differences even after 4 times lithography (**Figure 3h**), revealing robust ferroelectricity of the nonstoichiometric 2D $\text{Fe}_{1+\alpha}\text{Te}_2$ nanoflakes. The excellent stability may be attributed to the covalent bonding of the intercalated metal ions with the van der Waals layer, thereby enhancing the chemical coordination saturation of the chalcogen atoms and inhibiting the intrusion of oxygen and the degradation of the crystal structure.

Related discussions were added to the revised **Manuscript** and **Supplementary Information** as follows:

“The resulting nonstoichiometric 2D $\text{Cr}_{1+\sigma}\text{Se}_2$ and $\text{Ni}_{1+\delta}\text{Se}_2$ show an obvious piezoelectric response with d_{33} of 0.65 pm/V and 6.78 pm/V, respectively, while

$\text{Fe}_{1+\alpha}\text{Te}_2$ and $\text{Cu}_{1+\zeta}\text{S}_2$ demonstrate switchable spontaneous polarization along in-plane (IP) and out-of-plane (OOP) directions with superior ambient air stability up to 6 months and a Curie temperature above 523 K.”

“To further confirm the exotic ferroelectric behavior, the OOP polarization lithography was undertaken by applying ± 6 V DC voltage on $\text{Fe}_{1+\alpha}\text{Te}_2$ (white and black dotted lines in **Figure 3h**). Subsequent PFM scanning presents two contrast areas in the phase image, suggesting the large-area switchable spontaneous polarization of nonstoichiometric 2D $\text{Fe}_{1+\alpha}\text{Te}_2$ (up left in **Figure 3h**). After sequential electrical field writing of different patterns and conducting PFM imaging, the ferroelectric domains of $\text{Fe}_{1+\alpha}\text{Te}_2$ nanoflakes correspondingly switch, showing distinct signal differences even after 4 times of lithography (**Figure 3h**).”

“The multiple lithography of ferroelectric domains suggests rewritable and robust polarization of nonstoichiometric 2D $\text{Fe}_{1+\alpha}\text{Te}_2$ nanoflakes.”

“Remarkably, the nonstoichiometric 2D $\text{Fe}_{1+\alpha}\text{Te}_2$ nanoflakes used for PFM characterization in **Figure 3g-k** have been stored in an air environment for 6 months, demonstrating remarkable ferroelectric response compared with freshly prepared nanoflakes and suggesting superior stability of the ferroelectric phase (**Supplementary Figure 18**). The high stability of nonstoichiometric 2D TMDs nanoflakes can also be identified by the temperature-dependent Raman spectra of both $\text{Fe}_{1+\alpha}\text{Te}_2$ and $\text{Cu}_{1+\zeta}\text{S}_2$ nanoflakes (**Supplementary Figure 19**), indicating that both Curie temperatures are above 523 K. The excellent stability may be attributed to the covalent bonding of the intercalated metal ions with the van der Waals layer, thereby enhancing the chemical coordination saturation of the chalcogen atoms and inhibiting the intrusion of oxygen and the degradation of the crystal structure.”

“In addition, the covalent bonds formed by the intercalated metal atoms and chalcogen atoms also potentially enhance the phase and crystal stability of the nonstoichiometric 2D TMDs.”

Fig. 3 | Tuning piezoelectricity of nonstoichiometric $\text{Cr}_{1+\sigma}\text{Se}_2$ nanoflakes and emerging ferroelectricity in nonstoichiometric $\text{Fe}_{1+\alpha}\text{Te}_2$ nanoflakes. **a** OOP amplitude images (up), phase images (middle) and intrinsic amplitude images (bottom) of a nonstoichiometric $\text{Cr}_{1+\sigma}\text{Se}_2$ nanoflake under different drive voltages. **b** Corresponding OOP amplitude evolution curve with drive AC voltage. Orange, pink, blue and prasinous lines are assigned to the resonance-amplified OOP amplitude of the sample, resonance-amplified OOP amplitude of the substrate, original intrinsic OOP amplitude of the sample and linearly fitted intrinsic OOP amplitude of the sample as a function of drive AC voltage, respectively. **c** Evolution of resonance-amplified OOP amplitudes of nonstoichiometric $\text{Cr}_{1+\sigma}\text{Se}_2$ nanoflakes with different thicknesses. **d** Effective d_{33} of nonstoichiometric $\text{Cr}_{1+\sigma}\text{Se}_2$ nanoflakes with different values of σ . **e** Height image and different AC voltage-driven intrinsic amplitude images of a nonstoichiometric $\text{Cr}_{1+\sigma}\text{Se}_2$ nanoflake. **f** Corresponding intrinsic IP amplitude in **e** as a function of the drive AC voltage of substrate and nanoflake sample. **g** Local OOP ferroelectric switching spectra under DC off state of a nonstoichiometric $\text{Fe}_{1+\alpha}\text{Te}_2$ nanoflake. **h** Phase images of a nonstoichiometric $\text{Fe}_{1+\alpha}\text{Te}_2$ nanoflake after sequential lithography by using different patterns. **i** Local IP ferroelectric hysteresis loops under DC off state of a nonstoichiometric $\text{Fe}_{1+\alpha}\text{Te}_2$ nanoflake. The black and white dotted line

frames indicate +6 V and -6 V lithography area, respectively. **j-k** IP amplitude (**j**) and IP phase (**k**) images of a nonstoichiometric $\text{Fe}_{1+\alpha}\text{Te}_2$ nanoflake after electrical field writing along two perpendicular lines. The black line and white line indicate first +6 V and then -6 V lithography.

Supplementary Figure 18. (a-d) Local ferroelectric switching loops under DC off state (a), height image (b), OOP amplitude image (c) and OOP phase image (d) of a freshly prepared nonstoichiometric $\text{Fe}_{1+\alpha}\text{Te}_2$ nanoflake. The 10 V polarized lithography was performed on the nanoflake before PFM scanning. The relatively high coercive voltage and phase difference of less than 180° are attributed to surface charging and electrostatics induced by insufficiently grounded.

Supplementary Figure 19. (a-d) Temperature-dependent Raman spectra and Raman peak intensity of a nonstoichiometric Fe_{1+α}Te₂ nanoflake (a-b) and a nonstoichiometric Cu_{1+ζ}S₂ nanoflake (c-d). Both Raman peak intensity of Fe_{1+α}Te₂ and Cu_{1+ζ}S₂ nanoflakes present fluctuation in a small range during heating from room temperature to 523 K, indicating strong stability of ferroelectric phase with a Curie temperature above 523 K.

2, Figure 3a displays images of nonstoichiometric Cr_{1+σ}Se₂ nanoflake under positive drive voltages. It would be beneficial to also showcase the results under negative voltage. Additionally, in Figures 3f and 3g, it is necessary to exclude the influence of morphology on the intrinsic ferroelectric domain distribution.

Response: Thanks for your kind reminder. We are sorry for the ambiguity of drive voltage in the manuscript. The drive voltage used for piezoelectric response measurement and ferroelectric domain imaging is AC voltage, which is switched between positive and negative during PFM scanning to detect the piezoelectric vibration and phase change of the sample. In contrast, the drive voltage in the hysteresis loop or for the lithography is DC voltage to reverse polarization direction. We have already added related voltage labels on the Figures and plots (Figure 3, Supplementary Figure S3–18).

To rule out the influence of morphology on the intrinsic ferroelectric domain distribution, height images of each nanoflake used for ferroelectric domain imaging were supplemented. The independent phase and amplitude images relative to topography height images of the nonstoichiometric nanoflakes exclude pseudo-ferroelectricity originating from morphology or surface oxidation/damage (Supplementary Figure 15, Supplementary Figure 16c, Supplementary Figure 17 and Supplementary Figure 18b).

Relevant revisions are listed as follows:

“Especially, morphology-independent phase and amplitude images of nonstoichiometric nanoflakes exclude the pseudo-ferroelectricity originating from surface topography or oxidation/damage (Supplementary Figure 15, Supplementary Figure 16c, Supplementary Figure 17 and Supplementary Figure 18b).”

“The drive voltage used for piezoelectric response measurement and ferroelectric domain imaging is AC voltage, which is switched between positive and negative during PFM scanning to detect the piezoelectric vibration and phase change of the sample, while the drive voltage in the hysteresis loop or for the lithography is DC voltage to reverse polarization direction.”

Fig. 3 | Tuning piezoelectricity of nonstoichiometric $\text{Cr}_{1+\sigma}\text{Se}_2$ nanoflakes and emerging ferroelectricity in nonstoichiometric $\text{Fe}_{1+\alpha}\text{Te}_2$ nanoflakes. a OOP amplitude images (up), phase images (middle) and intrinsic amplitude images (bottom) of a nonstoichiometric $\text{Cr}_{1+\sigma}\text{Se}_2$ nanoflake under different drive voltages. **b** Corresponding OOP amplitude evolution curve with drive AC voltage. Orange, pink, blue and prasinous lines are assigned to the resonance-amplified OOP amplitude of the sample, resonance-amplified OOP amplitude of the substrate, original intrinsic OOP amplitude of the sample and linearly fitted intrinsic OOP amplitude of the sample as a

function of drive AC voltage, respectively. **c** Evolution of resonance-amplified OOP amplitudes of nonstoichiometric $\text{Cr}_{1+\sigma}\text{Se}_2$ nanoflakes with different thicknesses. **d** Effective d_{33} of nonstoichiometric $\text{Cr}_{1+\sigma}\text{Se}_2$ nanoflakes with different values of σ . **e** Height image and different AC voltage-driven intrinsic amplitude images of a nonstoichiometric $\text{Cr}_{1+\sigma}\text{Se}_2$ nanoflake. **f** Corresponding intrinsic IP amplitude in **e** as a function of the drive AC voltage of substrate and nanoflake sample. **g** Local OOP ferroelectric switching spectra under DC off state of a nonstoichiometric $\text{Fe}_{1+\alpha}\text{Te}_2$ nanoflake. **h** Phase images of a nonstoichiometric $\text{Fe}_{1+\alpha}\text{Te}_2$ nanoflake after sequential lithography by using different patterns. **i** Local IP ferroelectric hysteresis loops under DC off state of a nonstoichiometric $\text{Fe}_{1+\alpha}\text{Te}_2$ nanoflake. The black and white dotted line frames indicate +6 V and -6 V lithography area, respectively. **j-k** IP amplitude (**j**) and IP phase (**k**) images of a nonstoichiometric $\text{Fe}_{1+\alpha}\text{Te}_2$ nanoflake after electrical field writing along two perpendicular lines. The black line and white line indicate first +6 V and then -6 V lithography.

Supplementary Figure 16. (a-b) Ferroelectric switching spectra at DC off state of a $\text{Cu}_{1+\sigma}\text{S}_2$ nanoflake. (c-e) Height image, amplitude image and phase image of a $\text{Cu}_{1+\sigma}\text{S}_2$ nanoflake after lithography of ± 10 V rectangular-ambulatory-plane field.

Supplementary Figure 17. (a-d) High images of a nonstoichiometric $\text{Fe}_{1+\alpha}\text{Te}_2$ nanoflake after sequential lithography with different patterns. (e-h) Corresponding amplitude images of (a-d). (i) AFM height image of a nonstoichiometric $\text{Fe}_{1+\alpha}\text{Te}_2$ nanoflake on which IP ferroelectric lithography is performed.

Supplementary Figure 18. (a-d) Local ferroelectric switching loops under DC off state (a), height image (b), OOP amplitude image (c) and OOP phase image (d) of a freshly prepared nonstoichiometric $\text{Fe}_{1+\alpha}\text{Te}_2$ nanoflake. The 10 V polarized lithography was performed on the nanoflake before PFM scanning. The relatively high coercive voltage and phase difference of less than 180° are attributed to surface charging and electrostatics induced by insufficiently grounded.

3, The ferroelectric reversal observed in Figure 3e does not appear to be a complete 180° change. Additionally, it would be beneficial to provide an atomic-level explanation for the intrinsic reasons behind ferroelectric reversal.

Response: Thanks for your valuable comments and suggestions. The shift of the ferroelectric phase difference from 180° in the reversal process is attributed to local electrostatic and surface charging effects (Supplementary Figure 16 and Supplementary Figure 18). Our latest experiments can achieve a phase difference close to 180° and a smaller coercive voltage by using a more preferable grounding method (Figure 3g,i). The incomplete 180° flipping of the ferroelectric domains during lithography may be due to the dwelling time being too short during electrical field writing or the polarization direction of the lowest energy being tilted instead of absolutely in-plane or out-of-plane (*Nat. Commun.* 2023, 14, 2757).

The formation mechanism of nonstoichiometric 2D compounds is atomic-level interlayer metal intercalations between sandwich MX₂ layers and slight intralayer defects. The partially intercalated metal atoms covalently bond with chalcogen atoms in the bone MX₂ layer, forming completely and partially coordinated chalcogen atoms. The heterogeneous van der Waals force and covalent bonding interaction may cause fluctuations in the van der Waals layers, which are responsible for local interlayer sliding and metal atom displacement. The partial intercalation of metal atoms breaks the center equilibrium of positive and negative units and produces net electric dipole moments while suffering external mechanical stress. Besides, the sliding in the MX₂ layer can create an inequivalent between adjacent layers, producing the interlayer spontaneous alignment of electric dipoles along the OOP direction. The reduced potential energy barrier of interlayer sliding ensures the essential switching of the ferroelectric domain under an external electric field. In addition to interlayer sliding, the structural distortion induced by asymmetrical metal coordination and potential intralayer defects also accelerate the occurrence of lateral electric dipole and yield stable IP polarization. Switching of the external electrical field can control the lateral displacement of the intercalated metal ions and intralayer defects, thereby reversing the direction of IP spontaneous polarization in nonstoichiometric 2D TMDs.

Related discussions and FPM characterizations were added to the revised **Manuscript** as follows:

“The incomplete 180° flipping of the ferroelectric domains during lithography may be due to short dwelling time during electrical field writing or tilted polarization direction of the lowest energy instead of absolutely in-plane or out-of-plane²⁸.”

“Those detailed electron microscopic analyses distinctly uncover that the formation mechanism of nonstoichiometric 2D compounds is mostly attributed to atomic-level interlayer metal intercalations between sandwich MX₂ layers and potential slight intralayer defects. The partially intercalated metal atoms covalently bond with chalcogen atoms in the bone MX₂ layer, forming completely and partially coordinated chalcogen atoms. The heterogeneous van der Waals force and covalent bonding interaction may cause fluctuations in the van der Waals layers, which are responsible

for local interlayer sliding and metal atom displacement. The partial intercalation of metal atoms breaks the center equilibrium of positive and negative units, producing net electric dipole moments while suffering from external mechanical stress. Besides, the sliding in the MX_2 layer can create an inequivalent between adjacent layers, producing the interlayer spontaneous alignment of electric dipoles along the OOP direction. The reduced potential energy barrier of interlayer sliding ensures the essential switching of the ferroelectric domain under an external electric field^{28,60–62}. In addition to interlayer sliding, the structural distortion induced by asymmetrical metal coordination and potential intralayer defects also accelerate the occurrence of lateral electric dipole and yield stable IP polarization^{63–65}. Switching of the external electrical field can control the lateral displacement of the intercalated metal ions and intralayer defects, thereby reversing the direction of IP spontaneous polarization in nonstoichiometric 2D TMDs.”

“For lithography, ferroelectric hysteresis loop and electrical curve measurements, nonstoichiometric 2D nanoflakes were transferred on Cr/Au (5/50 nm) coated Si substrate and then grounded by connecting a conductive cable via the adhesion of silver paste.”

Fig. 3 | Tuning piezoelectricity of nonstoichiometric $\text{Cr}_{1+\sigma}\text{Se}_2$ nanoflakes and

emerging ferroelectricity in nonstoichiometric $\text{Fe}_{1+\alpha}\text{Te}_2$ nanoflakes. **a** OOP amplitude images (up), phase images (middle) and intrinsic amplitude images (bottom) of a nonstoichiometric $\text{Cr}_{1+\sigma}\text{Se}_2$ nanoflake under different drive voltages. **b** Corresponding OOP amplitude evolution curve with drive AC voltage. Orange, pink, blue and prasinous lines are assigned to the resonance-amplified OOP amplitude of the sample, resonance-amplified OOP amplitude of the substrate, original intrinsic OOP amplitude of the sample and linearly fitted intrinsic OOP amplitude of the sample as a function of drive AC voltage, respectively. **c** Evolution of resonance-amplified OOP amplitudes of nonstoichiometric $\text{Cr}_{1+\sigma}\text{Se}_2$ nanoflakes with different thicknesses. **d** Effective d_{33} of nonstoichiometric $\text{Cr}_{1+\sigma}\text{Se}_2$ nanoflakes with different values of σ . **e** Height image and different AC voltage-driven intrinsic amplitude images of a nonstoichiometric $\text{Cr}_{1+\sigma}\text{Se}_2$ nanoflake. **f** Corresponding intrinsic IP amplitude in **e** as a function of the drive AC voltage of substrate and nanoflake sample. **g** Local OOP ferroelectric switching spectra under DC off state of a nonstoichiometric $\text{Fe}_{1+\alpha}\text{Te}_2$ nanoflake. **h** Phase images of a nonstoichiometric $\text{Fe}_{1+\alpha}\text{Te}_2$ nanoflake after sequential lithography by using different patterns. **i** Local IP ferroelectric hysteresis loops under DC off state of a nonstoichiometric $\text{Fe}_{1+\alpha}\text{Te}_2$ nanoflake. The black and white dotted line frames indicate +6 V and -6 V lithography area, respectively. **j-k** IP amplitude (**j**) and IP phase (**k**) images of a nonstoichiometric $\text{Fe}_{1+\alpha}\text{Te}_2$ nanoflake after electrical field writing along two perpendicular lines. The black line and white line indicate first +6 V and then -6 V lithography.

4, Do other materials exhibit a similar trend in piezoelectric coefficients with changes in chemical composition as observed in $\text{Cr}_{1+\sigma}\text{Se}_2$? It would be advantageous to provide a more detailed and physics-based explanation of how variations in composition impact the structure and piezoelectric coefficients.

Response: Thanks for your insightful comments. We further conducted a comprehensive investigation on non-stoichiometric $\text{Ni}_{1+\delta}\text{Se}_2$ nanoflakes to study the evolution of piezoelectric coefficient d_{33} with chemical composition. We first prepared non-stoichiometric $\text{Ni}_{1+\delta}\text{Se}_2$ nanoflakes with various compositions using different chalcogen vapors and then determined the chemical ratio via EDS spectra (Supplementary Figure 9). Then, OOP PFM characterizations were performed on each kind of nanoflake, and the corresponding effective d_{33} was calculated by fitting the OOP intrinsic amplitude as a function of drive AC voltage (Supplementary Figure 10). A valley-like evolution curve was demonstrated by plotting the effective d_{33} and δ values (Supplementary Figure 11).

Then, we provided a more detailed and in-depth discussion on the relationship between crystal composition/structure and piezoelectric coefficients (Supplementary Note 3). We first discussed the relatively simple non-stoichiometric $\text{Cr}_{1+\sigma}\text{Se}_2$ model. The two terminals of the abscissa represent standard stoichiometric compounds of CrSe_2 and Cr_2Se_3 , which are centrosymmetry with a theoretical effective d_{33} value of zero. After including the two endpoints, the fitting curve depicts a mountain-like shape, indicating an evolution of first increase and then decrease of effective d_{33} as the σ value

increases (Figure 3d). The changes in piezoelectric coefficient can be associated with the variation of generated electric dipole density during applying identical mechanical stress, where the dipole density is a vector field consisting of individual dipole vectors. Assuming that each intercalated metal atom can produce a dipole for CrSe₂, while each intercalated metal atom vacancy can similarly generate an equivalent dipole in the opposite direction for Cr₂Se₃. Thus, the effective d_{33} will almost linearly increase with the number of interstitial defects (increase in the σ value of CrSe₂) or vacancy defects (decrease in the σ value of Cr₂Se₃). However, the effective d_{33} value will decrease after reaching the peak value due to the electrical dipole in the opposite direction canceling out, thereby inhibiting the enhancement of the net dipole density. Similar results are also observed in Ni_{1+ δ} Se₂ nanoflakes (Supplementary Figure 11b), while NiSe₂, Ni₃Se₄ and NiSe are used as a zero-value coordinate point. It is worth noting that the amplitude of the dipoles formed in an interstitial ion defect or a vacancy defect is not strictly equal, leading to a shift of peak position in the actual curve. Those result also highlights the significance of the nonstoichiometric ratio in tuning piezoelectricity in 2D materials.

Related revisions were made in the **Manuscript** and **Supplementary Information** as follows:

“The resulting nonstoichiometric 2D Cr_{1+ σ} Se₂ and Ni_{1+ δ} Se₂ show an obvious piezoelectric response with d_{33} of 0.65 pm/V and 6.78 pm/V, respectively, while Fe_{1+ α} Te₂ and Cu_{1+ ζ} S₂ demonstrate switchable spontaneous polarization along in-plane (IP) and out-of-plane (OOP) directions.”

“Thus, by fitting multiple points of intrinsic piezoelectric amplitude of the sample to the drive AC voltages, the effective d_{33} of a 5.6 nm thick nonstoichiometric 2D Cr_{1+ σ} Se₂ nanoflake and a 47.8 nm thick nonstoichiometric 2D Ni_{1+ δ} Se₂ were counted to be 0.65 \pm 0.12 pm/V and 6.78 \pm 0.6 pm/V, respectively (Figure 3b, more details are elaborated in Supplementary Note 2 and Supplementary Figure 5–11).”

“Especially, the thick nonstoichiometric 2D Ni_{1+ δ} Se₂ nanoflake shows superior piezoelectric performance, even comparable to well-known intrinsic 2D α -In₂Se₃ piezoelectrics (Supplementary Table 2)⁵⁹.”

“The chemical composition tuned Cr_{1+ σ} Se₂ and Ni_{1+ δ} Se₂ nanoflakes show a noticeable first decrease and then increase of effective d_{33} with the increase of σ/δ value (Figure 3d, Supplementary Figure 7–11, Supplementary Note 3). Therefore, the piezoelectric coefficient is more sensitive to the initial metal intercalation/vacancy. A small number of defects can rapidly enhance the piezoelectric coefficient, highlighting the significance of the nonstoichiometric ratio in engineering piezoelectricity in 2D materials.”

“The samples used for Supplementary Figure 9–11, Supplementary Figure 13–14 and Supplementary Figure 21 were prepared in the second batch, which is similar to the second batch preparation of Fe_{1+ α} Te₂. However, NiCl₂·6H₂O powder (35 mg, 99.9%, Sigma-Aldrich) and Se powder (3 g, 99.99%, Aladdin) were used as metal and chalcogen element precursors. The tube chamber was first purged by pure argon, and then the gas flow was maintained at 400 sccm Ar and 2 sccm H₂. The furnace zone was ramped to 200 °C over 30 min and then to 650 °C within 35 min, and subsequently held

for 10 min. The Se precursor was placed upstream at different distances from the edge of the furnace chamber, where the temperature at each position was recorded. After that, the furnace naturally cooled to room temperature.”

“Supplementary Note 3. The impacts of variation of chemical composition on the crystal structure and piezoelectric coefficient.

Defect/composition engineering was usually considered an effective/easy way to break the inversion symmetry of 2D materials to generate piezoelectricity and even spontaneous polarization. Thus, taking 2D $\text{Cr}_{1+\sigma}\text{Se}_2$ and $\text{Ni}_{1+\delta}\text{Se}_2$ nanoflake as examples, the σ/δ values of additional metal atoms were controlled by selenium vapor pressure/temperature and then determined by multi-point energy dispersive X-ray spectroscopy (EDS) (Supplementary Figure 7,9). The extra/intercalated metal atoms are verified by the positive value of σ/δ . Effective d_{33} of $\text{Cr}_{1+\sigma}\text{Se}_2$ nanoflake with the same thickness in different σ values were then determined by PFM characterization (Figure 3d and Supplementary Figure 8). Parallel experiments were also conducted on the $\text{Ni}_{1+\delta}\text{Se}_2$ systems (Supplementary Figure 9–11). As a result, both effective d_{33} of nonstoichiometric $\text{Cr}_{1+\sigma}\text{Se}_2$ and $\text{Ni}_{1+\delta}\text{Se}_2$ nanoflakes change with chemical composition (Figure 3d and Supplementary Figure 11).

The relatively simple non-stoichiometric $\text{Cr}_{1+\sigma}\text{Se}_2$ model is discussed first. The two terminals of the abscissa represent standard stoichiometric compounds of CrSe_2 and Cr_2Se_3 , which are centrosymmetry with a theoretical effective d_{33} value of zero. The fitting curve after including the two endpoints depicts a mountain-like shape, indicating an evolution of first increase and then decrease of effective d_{33} as the σ value increases (Figure 3d). The changes in piezoelectric coefficient can be associated with the variation of generated electric dipole density during applying identical mechanical stress, where the dipole density is a vector field consisting of individual dipole vectors. Assuming that each intercalated metal atom can produce a dipole for CrSe_2 , while each intercalated metal atom vacancy can similarly generate an equivalent dipole in the opposite direction for Cr_2Se_3 . Thus, the effective d_{33} will almost linearly increase with the number of interstitial defects (increase in the σ value of CrSe_2) or vacancy defects (decrease in the σ value of Cr_2Se_3). However, the effective d_{33} value will decrease after reaching the peak value due to the electrical dipole in the opposite direction canceling out, thereby inhibiting the enhancement of the net dipole density. Similar results are also observed in $\text{Ni}_{1+\delta}\text{Se}_2$ nanoflakes (Supplementary Figure 11b), while NiSe_2 , Ni_3Se_4 and NiSe are used as a zero-value coordinate point. It is worth noting that the amplitude of the dipoles formed in an interstitial ion defect or a vacancy defect is not strictly equal, leading to a shift of peak position in the actual curve.”

Supplementary Figure 9. (a-d) SEM image (upper left), EDS spectra (upper right) and chemical element ratio of four points marked in SEM image (bottom) of nonstoichiometric $\text{Ni}_{1+\delta}\text{Se}_2$ nanoflakes prepared under different chalcogen vapor pressures.

Supplementary Figure 10. (a-d) Height images (left), different drive AC voltages driven corresponding OOP intrinsic amplitude images (middle) and plots of OOP intrinsic sample/substrate amplitude as a function of drive AC voltage (right) of nonstoichiometric $\text{Ni}_{1+\delta}\text{Se}_2$ nanoflakes with different δ values. The δ values were determined by averaging the chemical ratio of four selected points in the SEM image (Supplementary Figure 9). According to the thickness-dependent piezoelectric coefficient evolution of nonstoichiometric $\text{Cr}_{1+\sigma}\text{Se}_2$ nanoflakes (Figure 3c), the piezoelectric coefficient is saturated when the thickness is higher than 20 nm. Thus, despite the slight difference in thickness, the piezoelectric coefficient of nonstoichiometric $\text{Ni}_{1+\delta}\text{Se}_2$ nanoflakes should be saturated for such thick nanoflakes, and the comparison is reasonable. The valid areas for counting sample amplitude were marked as light blue patterns in the last amplitude image. The effective d_{33} of each nanoflake with different δ values was calculated by fitting OOP intrinsic amplitude as a function of drive AC voltage.

Supplementary Figure 11. The effective d_{33} of nonstoichiometric $\text{Ni}_{1+\delta}\text{Se}_2$ nanoflakes as a function of δ value before (a) and after (b) considering centrosymmetric NiSe_2 and NiSe . The scatters were fitted via Polynomial fitting.

5, The authors are suggested to provide a comparison between the in-plane and out-of-plane results for the magnetic hysteresis loops shown in Figure 5e.

Response: Thanks for your kind suggestion. We conducted vibrating sample magnetometer (VSM) measurements of nonstoichiometric 2D $\text{Fe}_{1+\alpha}\text{Te}_2$ nanoflakes (Supplementary Figure 25). The magnetic hysteresis loops were observed at 10 K to 300 K temperatures in parallel and vertical magnetic fields, indicating the long-range and room-temperature magnetism order (Supplementary Figure 25). The saturation magnetization and coercive field of the IP field are higher than that of the OOP field, suggesting the easy axis direction is parallel to the van der Waals layer plane. The simultaneous existence of room-temperature magnetism and ferroelectricity in the

nonstoichiometric 2D $\text{Fe}_{1+\alpha}\text{Te}_2$ nanoflakes demonstrate the successful engineering of potential multiferroic material.

Related revisions were made as follows:

“As a verification, magnetic hysteresis loops are demonstrated in nonstoichiometric 2D $\text{Fe}_{1+\alpha}\text{Te}_2$ nanoflakes at 10 K and 300 K, indicating the simultaneous existence of room-temperature magnetism and ferroelectricity in the stoichiometrically engineered 2D TMDs (Supplementary Figure 25).”

Supplementary Figure 25. (a-b) M-H magnetization curves of nonstoichiometric 2D $\text{Fe}_{1+\alpha}\text{Te}_2$ nanoflake under parallel (IP) (a) and vertical (OOP) (b) magnetic fields. In the IP and OOP measurement configuration, the magnetic hysteresis loops were observed at 10 K and 300 K, indicating the long-range and room-temperature magnetism order. The saturation magnetization and coercive field measured under IP field are higher than that under the OOP field, suggesting the easy axis direction is parallel to the van der Waals layer plane.

Thanks again for your professional comments, which greatly help us improve the manuscript. Hopefully, we've addressed all of your concerns.

Reviewer #3 (Remarks to the Author):

In this work, Shu Ping Lau. et al proposed 2D novel metal intercalated $\text{Fe}_{1+\alpha}\text{Te}_2$, $\text{Co}_{1+\beta}\text{S}_2$, $\text{Mn}_{1+\gamma}\text{Se}_2$, $\text{Ni}_{1+\delta}\text{Se}_2$, $\text{V}_{1+\epsilon}\text{Se}_2$, $\text{Cu}_{1+\zeta}\text{S}_2$, and $\text{Cr}_{1+\sigma}\text{Se}_2$, in which the atomic-scale microstructure of interlayer metal atoms was revealed using STEM-iDPC. The piezoelectric performance of representative 2D TMDs was confirmed through PFM, while the dielectric or electrostatic effects or flexoelectricity were excluded. Their piezoelectric coefficients were found to be correlated with the thickness and chemical composition. The paper is well organized and the idea is novel. I recommend this high-quality work be published in Nature Communications after addressing the following two comments.

1. The out-of-plane (OP) piezoelectricity of nonstoichiometric 2D TMDs was strong, while the in-plane (IP) PFM mapping images showed weak and unstable piezoelectric responses (Supplementary Figure 5). In Figure 5, the authors demonstrated a familiar two-terminal flexible device to manifest its applications in nanogenerators, memristors, and synapses. The release and stretch processed both contain vertical and lateral deformation in a micro lateral size crystal, bringing vertical and lateral stress on nonstoichiometric 2D TMDs. According to the above PFM results, vertical direction stress (out-of-plane, d_{33}) contributed a large quantity of piezoelectricity-generated electrons. However, the electrodes of the two terminal devices were lateral direction (IP, $d_{11}/d_{22}/d_{21}$), indicating that the electrons mainly came from IP piezoelectricity (Figure 5a and 5f). Therefore, I suggest the authors fabricate an up-down two-terminal electrode device to be consistent with its OP piezoelectricity. Otherwise, the d_{31} direction piezoelectricity should be carried out and studied systematically.

Response: Thank you for the kind reminder. We deeply understand the inconsistency between the IP devices and OOP PFM characterizations. Thus, we have supplemented extra experiments to comprehensively investigate and clarify the IP and OOP piezo/ferroelectricity of non-stoichiometric 2D TMDs, including IP piezoelectric response of nonstoichiometric $\text{Cr}_{1+\sigma}\text{Se}_2$ nanoflake (Figure 3e,f) and $\text{Ni}_{1+\delta}\text{Se}_2$ nanoflakes (Supplementary Figure 13,14), and IP PFM hysteresis loops (Figure 3i), IP domain lithography (Figure 3j-k) and I-V electrical curve along vertical direction (OOP) (Figure 5e) of $\text{Fe}_{1+\alpha}\text{Te}_2$ nanoflake.

Lateral PFM (IP mode) can detect the in-plane component of piezoelectric response as the lateral motion of the cantilever due to bias-induced surface shearing. As the drive voltage is just applied on the tips, the calculated piezoelectric coefficient is referred to d_{31}/d_{32} . After calibrating by standard sample such as a z-cut lithium niobate wafer, it is completely sufficient to qualitatively reveal the IP ferroelectric domains and calculate IP piezoelectric coefficients.

Our further experiments found that the high-resolution IP PFM characterizations of $\text{Cr}_{1+\sigma}\text{Se}_2$ and $\text{Ni}_{1+\delta}\text{Se}_2$ nanoflakes can more distinctly reveal the evolution of IP intrinsic amplitude relevant to drive AC voltage (Figure 3e,f and Supplementary Figure 13). To

accurately describe the IP piezoelectric coefficient, it is crucial to detect the angle-resolved IP piezoelectric response to specify the contribution of d_{31} and d_{32} . Thus, IP PFM characterizations were performed on three $\text{Ni}_{1+\delta}\text{Se}_2$ nanoflakes with different alignment directions (Supplementary Figure 14). As the six-fold symmetry of the hexagonal crystals, the nearly identical IP piezoelectric coefficients of the three nanoflakes are sufficient to identify the isotropy of the IP piezoelectric response of the nonstoichiometric 2D nanoflakes (Supplementary Figure 14). Thus, after calibrated by z -cut lithium niobate (LN) crystals, effective $d_{31/32}$ of $\text{Cr}_{1+\sigma}\text{Se}_2$ and $\text{Ni}_{1+\delta}\text{Se}_2$ nanoflake were calculated to be 0.12 ± 0.005 pm/V and 0.16 ± 0.017 pm/V, respectively (Figure 3e,f and Supplementary Figure 13). The IP piezoelectric coefficient is not too strong, but it can sufficiently indicate essential IP piezoelectric response in nonstoichiometric nanoflakes and application in nanogenerators.

Moreover, IP ferroelectric switching spectra (Figure 3i) and ferroelectric domain lithography (Figure 3j-k) were also performed to conform to the lateral two-terminal memristor. The corresponding IP switching spectroscopy shows typical hysteresis behavior with a loop window (Figure 3i). Besides, the switching of IP spontaneous polaritons of nonstoichiometric 2D $\text{Fe}_{1+\alpha}\text{Te}_2$ nanoflakes was also verified by sequentially scanning black line with 6 V and white line with -6 V voltage along the direction of arrow in Figure 3j,k. The first lithography with a positive voltage (black line) shows obvious manipulation of the polarization direction (Supplementary Note 4, Figure 3k), and subsequent negative voltage line, in turn, changes the direction of ferroelectric polarization (as displayed in the intersection in Figure 3k). The manifestation of IP ferroelectricity in non-stoichiometric $\text{Fe}_{1+\sigma}\text{Te}_2$ nanoflakes suggests it is a candidate material for in-plane ferroelectric memory devices.

In addition, we collected the I-V curves along the vertical (out-of-plane) direction of the nanoflake to study electrical behavior corresponding to OOP ferroelectricity. The I-V curves of the vertical device show typical loop windows at different ranges of DC voltage, confirming the reversal of the polarization states at more negative or positive voltages and thus changing the resistance of the nanoflakes (Figure 5e). As the voltage range increases, the loop window accordingly expands. The I-V characteristics substantially verify the OOP ferroelectricity, signifying the FTJ application (Figure 5e). However, higher voltage also potentially enhances the tunneling current and causes the loop deviation (as indicated by the I-V curve with a sweeping range of -10 V to 10 V in Figure 5e).

Thus, we proposed a logical understanding between the performance of lateral two-terminal nanogenerator and IP piezoelectric response. Besides, a one-to-one logical relationship between ferroelectricity and ferroelectric-based applications in IP and OOP modes was also established.

Relevant revisions have been made as follows:

“The high-resolution IP PFM characterizations of $\text{Cr}_{1+\sigma}\text{Se}_2$ and $\text{Ni}_{1+\delta}\text{Se}_2$ nanoflakes can more distinctly reveal the evolution of IP intrinsic amplitude relevant to drive AC voltage (Figure 3e,f and Supplementary Figure 13). To accurately describe the IP piezoelectric coefficient, it is crucial to detect the angle-resolved IP piezoelectric

response to specify the contribution of d_{31} and d_{32} . Thus, IP PFM characterizations were performed on three $\text{Ni}_{1+\delta}\text{Se}_2$ nanoflakes with different alignment directions (Supplementary Figure 14). As the six-fold symmetry of the hexagonal crystals, the nearly identical IP piezoelectric coefficients of the three nanoflakes are sufficient to identify the isotropy of the IP piezoelectric response of the nonstoichiometric 2D nanoflakes (Supplementary Figure 14). Thus, after calibrated by z -cut lithium niobate (LN) crystals, effective $d_{31/32}$ of $\text{Cr}_{1+\delta}\text{Se}_2$ and $\text{Ni}_{1+\delta}\text{Se}_2$ nanoflake were calculated to be 0.12 ± 0.005 pm/V and 0.16 ± 0.017 pm/V, respectively (Figure 3e,f and Supplementary Figure 13). The IP piezoelectric coefficient is not very impressive, but is sufficient to indicate essential IP piezoelectric response in nonstoichiometric nanoflakes and application in nanogenerators.”

“Besides, the switching of IP spontaneous polaritons of nonstoichiometric 2D $\text{Fe}_{1+\alpha}\text{Te}_2$ nanoflakes was also verified by sequentially scanning a black line with 6 V and a white line with -6 V voltage along the direction of the arrow in Figure 3j,k. The first lithography with a positive voltage (black line) shows obvious manipulation of the polarization direction (Supplementary Note 4, Figure 3k), and subsequent negative voltage line, in turn, changes the direction of ferroelectric polarization (as displayed in the intersection in Figure 3k). The multiple lithography of ferroelectric domains suggests rewritable and robust polarization of nonstoichiometric 2D $\text{Fe}_{1+\alpha}\text{Te}_2$ nanoflakes.”

“The IP and OOP ferroelectricity of the 2D $\text{Fe}_{1+\alpha}\text{Te}_2$ nanoflakes make them candidates for the ferroelectric tunnel junction (FTJ) and ferroelectric memristor applications. Firstly, the vertical electrical transport behaviors were investigated to elaborate OOP ferroelectricity via conductive force microscopy (CFM) (Figure 5e). The high voltages (more negative or positive) reverse the polarization direction of the ferroelectric nanoflakes, leading to the two different resistances associated with different I-V curve slopes (Figure 5e). The enlarging loop windows with the increase of voltage range substantially verify the OOP ferroelectricity and signify the FTJ application (Figure 5e). However, higher voltage also potentially enhances the tunneling current and causes the loop deviation (as indicated by the I-V curve with a sweeping range of -10 V to 10 V in Figure 5e).”

“Supplementary Note 2. Definition and calculation of effective piezoelectric coefficient d_{33} and $d_{31/32}$.

2.1 Definition of d_{33} and $d_{31/32}$

Piezoelectricity is electric polarization in a crystal caused by external stress. The piezoelectric effect can be described by $P_i = d_{ij}\sigma_j$, where P is polarization, d is a direct piezoelectric coefficient, σ is applied stress, and i and j imply polarization direction and stress direction. PFM is based on the inverse piezoelectric effect, where the applied voltage will induce sample deformation. The inverse piezoelectric effect can be expressed as $\varepsilon_j = d_{ij}E_i$, in which ε_j is the deformation produced by an electric field component E_i . In our experiment, drive voltages were applied along the vertical direction while IP and OOP piezoelectric amplitude and phase were measured, resulting in a calculated piezoelectric coefficient of effective d_{33} and $d_{31/32}$. The so-called

effective d_{33} and $d_{31/32}$ are because the measured value depends on the crystal topography, tip-sample contact condition, real space direction, and so on.

2.1 Calculation of effective d_{33} and $d_{31/32}$

A single frequency PFM mode was used to reveal in-plane piezoelectric response (Supplementary Figure 12), while dual AC resonance tracking (DART) PFM with lateral and vertical modes were used to measure IP and OOP ferroelectric domains, piezoelectric response and corresponding piezoelectric coefficient. In the DART OOP and IP modes, two frequencies near the center of the intrinsic resonance peak were chosen to collect two sets of amplitude and phase images (Supplementary Figure 3). However, those amplitudes are amplified by resonance rather than real/intrinsic amplitudes. The real/intrinsic amplitudes are then calculated by a simple harmonic oscillator model (SHO) (such as the bottom planes in Figure 3a and 3e). The grey spots indicate missing data which is unsuccessfully calculated during SHO calculation. The mean amplitude value and corresponding standard deviation for the interested region in the sample or substrate are exported by performing a Gaussian fit on the statistical diagram of all site values of the amplitude image. By linear fitting intrinsic amplitude to the drive AC voltage under consideration of the error bars, the mean value of effective piezoelectric coefficient (slope) and standard deviation are recorded (Figure 3b,f and Supplementary Figure 5,10,13). The value of the effective IP piezoelectric coefficient ($d_{31/32}$) is finally calibrated by a standard z -cut LN crystal.

A detailed region-cutting method was conducted to take account of the spatial variation of PFM amplitudes and evaluate the deviation of the piezoelectric coefficient in space. As illustrated in Supplementary Figure 5b,c, the intrinsic amplitude image of nanoflake in Supplementary Figure 5a acquired at a drive AC voltage of 4 V was divided into three regions, and the corresponding effective d_{33} was separately calculated through the method mentioned above. The calculated effective d_{33} values of area I, II and III were 0.61 ± 0.12 pm/V, 0.67 ± 0.14 pm/V and 0.63 ± 0.10 pm/V, respectively. The value differences between the three effective d_{33} are only in the second significant digit and are within the margin of standard error. Moreover, the average value of the effective d_{33} of three different regions is about 0.64 pm/V, which is very close to the effective d_{33} (0.65 ± 0.12 pm/V) counted from the total blue region (Supplementary Figure 5a and Figure 3b). Although the local non-uniform distribution of amplitude signals will produce different effective d_{33} , the effective d_{33} calculated from the signal in the entire nanoflake region can be used to represent the piezoelectric coefficient of the nanoflakes.”

“Supplementary Note 4. Elaboration of IP PFM characterization and lithography.

Lateral PFM (IP mode) can detect an in-plane component of the dipoles as the lateral motion of the cantilever due to bias-induced surface shearing. Owing to the small contact area of the conductive tip on which drive AC voltage is applied, the revealed IP domains may not be as robust as the OOP one and the calculated effective piezoelectric coefficient is related to d_{31}/d_{32} . However, it is completely sufficient to qualitatively reveal the IP ferroelectric domains and calculate piezoelectric coefficients with reference to standard z -cut lithium niobate (LN) wafer. Moreover, angle-dependent IP PFM were performed to precisely ascertain the d_{31} and d_{32} component of IP

piezoelectric response (Supplementary Figure 14).

Unlike switching the ferroelectric domain through patterns in the OOP lithography, two perpendicular scan lines with opposite voltages are employed to switch IP ferroelectric domains to avoid interaction between each scan line in the pattern (Figure 3j,k). Since the applied DC voltage is along the vertical direction, the electric field is a sphere centered on the tip. Besides, the diameter of the tip is very small, with tens of nanometers. Thus, the dwell time of each point in the line scan is prolonged compared with pattern lithography. After line scan lithography, the polarization directions on both sides of the scan line are theoretically opposite and perpendicular to the polarization direction on the scan line path. However, the IP PFM can only detect the polariton components perpendicular to the tip cantilever. Therefore, the phase patterning along the transverse line (black line) is obvious, while the phase along the longitudinal direction shows the same degree as the two side areas after lithography (white line) (Figure 3k). However, the horizontal polarization direction is obviously changed after longitudinal scanning (as shown by the point of intersection in Figure 3k), strongly proving the reverse of IP polarization of nonstoichiometric 2D TMDs. In addition, the phase and amplitude of the substrate are almost unchanged after line lithography, demonstrating the phase change originates from the reversion of the IP polarization rather than the charge accumulation effect.

”

Fig. 3 | Tuning piezoelectricity of nonstoichiometric $\text{Cr}_{1+\sigma}\text{Se}_2$ nanoflakes and emerging ferroelectricity in nonstoichiometric $\text{Fe}_{1+\alpha}\text{Te}_2$ nanoflakes. **a** OOP amplitude images (up), phase images (middle) and intrinsic amplitude images (bottom) of a nonstoichiometric $\text{Cr}_{1+\sigma}\text{Se}_2$ nanoflake under different drive voltages. **b** Corresponding OOP amplitude evolution curve with drive AC voltage. Orange, pink, blue and prasinous lines are assigned to the resonance-amplified OOP amplitude of the sample, resonance-amplified OOP amplitude of the substrate, original intrinsic OOP amplitude of the sample and linearly fitted intrinsic OOP amplitude of the sample as a function of drive AC voltage, respectively. **c** Evolution of resonance-amplified OOP amplitudes of nonstoichiometric $\text{Cr}_{1+\sigma}\text{Se}_2$ nanoflakes with different thicknesses. **d** Effective d_{33} of nonstoichiometric $\text{Cr}_{1+\sigma}\text{Se}_2$ nanoflakes with different values of σ . **e** Height image and different AC voltage-driven intrinsic amplitude images of a nonstoichiometric $\text{Cr}_{1+\sigma}\text{Se}_2$ nanoflake. **f** Corresponding intrinsic IP amplitude in **e** as a function of the drive AC voltage of substrate and nanoflake sample. **g** Local OOP ferroelectric switching spectra under DC off state of a nonstoichiometric $\text{Fe}_{1+\alpha}\text{Te}_2$ nanoflake. **h** Phase images of a nonstoichiometric $\text{Fe}_{1+\alpha}\text{Te}_2$ nanoflake after sequential lithography by using different patterns. **i** Local IP ferroelectric hysteresis loops under DC off state of a nonstoichiometric $\text{Fe}_{1+\alpha}\text{Te}_2$ nanoflake. The black and white dotted line

frames indicate +6 V and -6 V lithography area, respectively. **j-k** IP amplitude (**j**) and IP phase (**k**) images of a nonstoichiometric $\text{Fe}_{1+\alpha}\text{Te}_2$ nanoflake after electrical field writing along two perpendicular lines. The black line and white line indicate first +6 V and then -6 V lithography.

Supplementary Figure 13. (a) Height image of a nonstoichiometric $\text{Ni}_{1+\delta}\text{Se}_2$ nanoflake. (b-e) Corresponding IP intrinsic amplitude images under different drive AC voltages. The δ of this nanoflake is 1.66 and the thickness of the nanoflake is about 63.3 nm to ensure the saturation of the piezoelectric coefficient. (f) The IP intrinsic amplitude as a function of drive AC voltage. The IP intrinsic amplitude is typically varied with the drive AC voltage, indicating the obvious IP piezoelectric response of the nonstoichiometric $\text{Ni}_{1+\delta}\text{Se}_2$ nanoflakes. The calculated effective $d_{31/32}$ is ~ 0.16 pm/V after calibrated by z -cut LN crystal.

Supplementary Figure 14. (a-f) Height images (top) and corresponding IP intrinsic amplitude images (bottom) of three nonstoichiometric $\text{Ni}_{1+\delta}\text{Se}_2$ nanoflakes with the

same thickness and chemical composition but different alignment directions. The piezoelectric coefficients in (d-f) were calculated by directly dividing the mean value of statistical amplitude by the drive AC voltage. The alignment direction is uniformly defined as the degree difference between the diagonal dotted line and the vertical reference straight line.

Fig. 5 | Electrical behavior of two-terminal devices based on nonstoichiometric 2D materials. **a** Digit photo images and an optical image of a two-terminal nanogenerator based on a nonstoichiometric 2D $\text{Cr}_{1+\sigma}\text{Se}_2$ nanoflake. **b** Output voltage and short circuit current of a 2D $\text{Cr}_{1+\sigma}\text{Se}_2$ nanoflake under one release-stretch-release cycle. **c-d** Output voltage and short circuit current as a function of time under several cycles of tensile strain. **e** Electrical transport behaviors of the nonstoichiometric 2D $\text{Fe}_{1+\alpha}\text{Te}_2$ nanoflakes along OOP direction. The inset illustrates the principal scheme of a CFM using a conductive tip and Au layer as top and bottom electrodes, respectively. **f** I-V curves of a two-terminal nonstoichiometric 2D $\text{Fe}_{1+\alpha}\text{Te}_2$ nanoflake device with different maximum sweeping voltages. **g** I-V curves of a two-terminal nonstoichiometric 2D $\text{Fe}_{1+\alpha}\text{Te}_2$ nanoflake device with 109 cycle sweeping. The sweeping voltage is set in a range of -4 to 4 V. **h** Drain currents retention curves read at drain voltages of 10 and 0.5 V (up) and read at 2 V after 10 V and -10 V bias poling (below). **i** I-V curves after 7 times accumulation. The inset shows the enlarged grey dotted frame of the curves in logarithmic coordinates. Drain voltage sweeps from 0 V to 10 V in each cycle. **j** Currents and open circuit voltages evolution in the 7 cycles sweeping.

2. In Supplementary Table 2, the d_{33} of nonstoichiometric 2D $\text{Cr}_{1+\sigma}\text{Se}_2$ is 0.71 ± 0.02 pm/V. Compared with other reported materials, the performance is not outstanding. Is there room for improvement?

Response: Thanks for your insightful question. The piezoelectric coefficient of nonstoichiometric compounds $\text{Cr}_{1+\sigma}\text{Se}_2$ and $\text{Ni}_{1+\delta}\text{Se}_2$ is mainly controlled by adjusting the chemical composition and thickness of the nanoflakes (Figure 3c,d and Supplementary Figure 11). After considering centrosymmetric standard stoichiometric compounds, the chemical composition-tuned $\text{Cr}_{1+\sigma}\text{Se}_2$ and $\text{Ni}_{1+\delta}\text{Se}_2$ nanoflakes show that the effective d_{33} first noticeably increases and then decreases the increase of σ/δ value (Figure 3d, Supplementary Figure 7–9, Supplementary Note 3). Besides, the piezoelectric coefficient is also largely tuned by the thickness (Figure 3c). Thus, to achieve a considerable effective d_{33} , it is necessary to ensure a certain thick thickness and simultaneously tune the proportion of intercalated metal atoms to close to the peak piezoelectric coefficient. However, for chromium-based 2D materials, the stoichiometric ratio of the thicker sample is closer to the standard value, and it is difficult to achieve the chemical ratio of peak piezoelectric coefficient in thick nanoflakes, which is mainly affected by the binding ability of the metal atom to the chalcogen atom. The highest d_{33} of $\text{Cr}_{1+\sigma}\text{Se}_2$ nanoflakes we have obtained so far is 0.65 ± 0.12 pm/V after considering the standard deviation of amplitude. However, our newest experiment found that the nonstoichiometric ratio of $\text{Ni}_{1+\delta}\text{Se}_2$ nanoflakes can be well maintained and tuned in thick nanoflakes, obtaining a very considerable d_{33} value of 6.78 ± 0.6 pm/V (Supplementary Figure 11). This value is comparable to well-known intrinsic 2D $\alpha\text{-In}_2\text{Se}_3$ piezoelectrics (Supplementary Table 2), suggesting a great potential of nonstoichiometric compounds in obtaining large piezoelectric coefficients.

Beyond this, another highlight of our work may be the flexible integration of piezo/ferroelectricity on 2D materials. 2D materials have been verified to possess many fascinating optical, electrical and mechanical properties. Engineering 2D materials with piezo/ferroelectricity have very important prospects for the development of new multifunctional devices. However, the challenge is the center-symmetrical crystal structure required for piezoelectricity. Our nonstoichiometric ratio engineering has achieved this ingenious combination and is expected to develop functional materials with various properties, such as multiferroics and magneto-electro-mechanic-coupled materials. We believe that our findings will be of great help and benefit to readership and researchers.

Relevant discussions and experiments have been supplemented in the **Manuscript** and **Supplementary Information** as follows:

“The resulting nonstoichiometric 2D $\text{Cr}_{1+\sigma}\text{Se}_2$ and $\text{Ni}_{1+\delta}\text{Se}_2$ show an obvious piezoelectric response with d_{33} of 0.65 pm/V and 6.78 pm/V, respectively, while $\text{Fe}_{1+\alpha}\text{Te}_2$ and $\text{Cu}_{1+\zeta}\text{S}_2$ demonstrate switchable spontaneous polarization along in-plane (IP) and out-of-plane (OOP) directions.”

“Thus, by fitting multiple points of intrinsic piezoelectric amplitude of the sample to

the drive AC voltages, the effective d_{33} of a 5.6 nm thick nonstoichiometric 2D $\text{Cr}_{1+\sigma}\text{Se}_2$ nanoflake and a 47.8 nm thick nonstoichiometric 2D $\text{Ni}_{1+\delta}\text{Se}_2$ were counted to be 0.65 ± 0.12 pm/V and 6.78 ± 0.6 pm/V, respectively (Figure 3b, more details are elaborated in Supplementary Note 2 and Supplementary Figure 5–11).”

“Especially, the thick nonstoichiometric 2D $\text{Ni}_{1+\delta}\text{Se}_2$ nanoflake shows superior piezoelectric performance, even comparable to well-known intrinsic 2D $\alpha\text{-In}_2\text{Se}_3$ piezoelectrics (Supplementary Table 2)⁵⁹.”

“The chemical composition tuned $\text{Cr}_{1+\sigma}\text{Se}_2$ and $\text{Ni}_{1+\delta}\text{Se}_2$ nanoflakes show a noticeable first decrease and then increase of effective d_{33} with the increase of σ/δ value (Figure 3d, Supplementary Figure 7–11, Supplementary Note 3). Therefore, the piezoelectric coefficient is more sensitive to the initial metal intercalation/vacancy. A small number of defects can rapidly enhance the piezoelectric coefficient, highlighting the significance of the nonstoichiometric ratio in engineering piezoelectricity in 2D materials.”

“Supplementary Note 3. The impacts of variation of chemical composition on the crystal structure and piezoelectric coefficient.

Defect/composition engineering was usually considered an effective/easy way to break the inversion symmetry of 2D materials to generate piezoelectricity and even spontaneous polarization. Thus, taking 2D $\text{Cr}_{1+\sigma}\text{Se}_2$ and $\text{Ni}_{1+\delta}\text{Se}_2$ nanoflake as examples, the σ/δ values of additional metal atoms were controlled by selenium vapor pressure/temperature and then determined by multi-point energy dispersive X-ray spectroscopy (EDS) (Supplementary Figure 7,9). The extra/intercalated metal atoms are verified by the positive value of σ/δ . Effective d_{33} of $\text{Cr}_{1+\sigma}\text{Se}_2$ nanoflake with the same thickness in different σ values were then determined by PFM characterization (Figure 3d and Supplementary Figure 8). Parallel experiments were also conducted on the $\text{Ni}_{1+\delta}\text{Se}_2$ systems (Supplementary Figure 9–11). As a result, both effective d_{33} of nonstoichiometric $\text{Cr}_{1+\sigma}\text{Se}_2$ and $\text{Ni}_{1+\delta}\text{Se}_2$ nanoflakes change with chemical composition (Figure 3d and Supplementary Figure 11).

The relatively simple non-stoichiometric $\text{Cr}_{1+\sigma}\text{Se}_2$ model is discussed first. The two terminals of the abscissa represent standard stoichiometric compounds of CrSe_2 and Cr_2Se_3 , which are centrosymmetry with a theoretical effective d_{33} value of zero. The fitting curve after including the two endpoints depicts a mountain-like shape, indicating an evolution of first increase and then decrease of effective d_{33} as the σ value increases (Figure 3d). The changes in piezoelectric coefficient can be associated with the variation of generated electric dipole density during applying identical mechanical stress, where the dipole density is a vector field consisting of individual dipole vectors. Assuming that each intercalated metal atom can produce a dipole for CrSe_2 , while each intercalated metal atom vacancy can similarly generate an equivalent dipole in the opposite direction for Cr_2Se_3 . Thus, the effective d_{33} will almost linearly increase with the number of interstitial defects (increase in the σ value of CrSe_2) or vacancy defects (decrease in the σ value of Cr_2Se_3). However, the effective d_{33} value will decrease after reaching the peak value due to the electrical dipole in the opposite direction canceling out, thereby inhibiting the enhancement of the net dipole density. Similar results are also observed in $\text{Ni}_{1+\delta}\text{Se}_2$ nanoflakes (Supplementary Figure 11b), while NiSe_2 , Ni_3Se_4

and NiSe are used as a zero-value coordinate point. It is worth noting that the amplitude of the dipoles formed in an interstitial ion defect or a vacancy defect is not strictly equal, leading to a shift of peak position in the actual curve.”

Supplementary Table 2. Effective d_{33} comparison between nonstoichiometric 2D $\text{Cr}_{1+\delta}\text{Se}_2$ nanoflake and other 2D piezoelectrics. The intrinsic represents the inherent non-centrosymmetric structure with piezoelectricity, and the extrinsic represents no piezoelectric effect in perfect crystals and the piezoelectricity obtained by breaking the central symmetric structure.

Materials	Origin of piezoelectric	Effective (pm/V)	d_{33}	Thickness (nm)	Ref.
SnS₂	Extrinsic	2 ± 0.22	4	4	3
MoO₂	Extrinsic	0.56	8.7	8.7	4
α-Tellurene	Intrinsic	1	2	2	5
CdS	Intrinsic	16.4	3.09	3.09	6
α-In₂Se₃	Intrinsic	5.6	Bulk	Bulk	7
		0.34	Monolayer		
3R-MoS₂	Intrinsic	≈ 0.9	18	18	8
2H-MoTe₂	Extrinsic	2.08	125	125	9
MoS₂/WS₂	Intrinsic	1.95 – 2.09	>1.5	>1.5	10
Janus MoSSe	Intrinsic	0.1	0.7	0.7	11
Doped graphene	Extrinsic	1.4	Monolayer	Monolayer	12
2D Cr_{1+δ}Se₂	Extrinsic	0.65 ± 0.12	5.6	5.6	This work
2D Ni_{1+δ}Se₂	Extrinsic	6.78 ± 0.6	47.8	47.8	This work

Supplementary Figure 9. (a-d) SEM image (upper left), EDS spectra (upper right) and chemical element ratio of four points marked in SEM image (bottom) of nonstoichiometric $\text{Ni}_{1+\delta}\text{Se}_2$ nanoflakes prepared under different chalcogen vapor pressures.

Supplementary Figure 10. (a-d) Height images (left), different drive AC voltages driven corresponding OOP intrinsic amplitude images (middle) and plots of OOP intrinsic sample/substrate amplitude as a function of drive AC voltage (right) of nonstoichiometric $\text{Ni}_{1+\delta}\text{Se}_2$ nanoflakes with different δ values. The δ values were determined by averaging the chemical ratio of four selected points in the SEM image (Supplementary Figure 9). According to the thickness-dependent piezoelectric coefficient evolution of nonstoichiometric $\text{Cr}_{1+\sigma}\text{Se}_2$ nanoflakes (Figure 3c), the piezoelectric coefficient is saturated when the thickness is higher than 20 nm. Thus, despite the slight difference in thickness, the piezoelectric coefficient of nonstoichiometric $\text{Ni}_{1+\delta}\text{Se}_2$ nanoflakes should be saturated for such thick nanoflakes, and the comparison is reasonable. The valid areas for counting sample amplitude were marked as light blue patterns in the last amplitude image. The effective d_{33} of each nanoflake with different δ values was calculated by fitting OOP intrinsic amplitude as a function of drive AC voltage.

Supplementary Figure 11. The effective d_{33} of nonstoichiometric $\text{Ni}_{1+\delta}\text{Se}_2$ nanoflakes as a function of δ value before (a) and after (b) considering centrosymmetric NiSe_2 and NiSe . The scatters were fitted via Polynomial fitting.

Thanks again for your professional comments, which greatly help us improve the manuscript. Hopefully, we've addressed all of your concerns.

REVIEWERS' COMMENTS

Reviewer #1 (Remarks to the Author):

The authors have addressed my questions. The revised manuscript has been improved. I would recommend the publication.

Reviewer #2 (Remarks to the Author):

The author has responded to my questions appropriately and made suitable modifications to the article. Therefore, I recommend its publication in Nature Communications.

Reviewer #3 (Remarks to the Author):

All my questions were addressed. I therefore recommend to publish it as it is.

Reviewer #1 (Remarks to the Author):

The authors have addressed my questions. The revised manuscript has been improved. I would recommend the publication.

Response: We sincerely appreciate your time in reviewing our manuscript. Your valuable insights have greatly enriched this work.

Reviewer #2 (Remarks to the Author):

The author has responded to my questions appropriately and made suitable modifications to the article. Therefore, I recommend its publication in Nature Communications.

Response: Thanks for your positive feedback and your support on publishing. Your constructive comments improve the quality of our work a lot.

Reviewer #3 (Remarks to the Author):

All my questions were addressed. I therefore recommend to publish it as it is.

Response: We express our sincere gratitude for your approval and support of this manuscript.